# LSD1 drives intestinal epithelial maturation and controls small intestinal immune cell composition independent of microbiota in a murine model

Alberto Díez-Sánchez [1] ✉, Håvard T. Lindholm [1,2], Pia M. Vornewald [1], Jenny Ostrop [1], Rouan Yao [1], Andrew B. Single [1], Anne Marstad[1], Naveen Parmar [1], Tovah N. Shaw [3], Mara Martín-Alonso [1] & Menno J. Oudhoff [1,4] ✉

Postnatal development of the gastrointestinal tract involves the establishment of the commensal microbiota, the acquisition of immune tolerance via a balanced immune cell composition, and maturation of the intestinal epithelium. While studies have uncovered an interplay between the first two, less is known about the role of the maturing epithelium. Here we show that intestinal-epithelial intrinsic expression of lysine-specific demethylase 1A (LSD1) is necessary for the postnatal maturation of intestinal epithelium and maintenance of this developed state during adulthood. Using microbiota-depleted mice, we find plasma cells, innate lymphoid cells (ILCs), and a specific myeloid population to depend on LSD1-controlled epithelial maturation. We propose that LSD1 controls the expression of epithelial-derived chemokines, such as *Cxcl16*, and that this is a mode of action for this epithelial-immune cell interplay in local ILC2s but not ILC3s. Together, our findings suggest that the maturing epithelium plays a dominant role in regulating the local immune cell composition, thereby contributing to gut homeostasis.

Early in life, after birth, the gastrointestinal tract must adapt to a changing environment. There is a nutritional switch from milk to solid foods, while at the same time, the microbiota shifts from a few pioneering species in low abundance towards a complex microbial ecosystem with a large load[1]. In addition, during this time, the immune system is educated so it provides tolerance to harmless foreign antigens, including commensal microbes[2,3]. Simultaneously, residing in between the microbiota and the immune system, a single layer of intestinal epithelial cells undergoes maturation, which includes the appearance of Paneth cells and the progression of other intestinal

epithelial cell types into an "adult" state[4,5]. Appropriate postnatal gastrointestinal adaptation is important for our wellbeing. Indeed, necrotizing enterocolitis remains a life-threatening disease for preterm infants[6], and fetal and early-life antibiotic regimens correlate with developing asthma later in life[7]. In general, gut physiology has been linked to an ever-increasing list of local and systemic diseases ranging from inflammatory bowel disease (IBD), to asthma, and even Alzheimer's disease[8].

The small intestinal epithelium dramatically changes in development. Immature villus structures originate at the fetal stage while

[1]Centre of Molecular Inflammation Research, Department of Clinical and Molecular Medicine, Norwegian University of Science and Technology, Trondheim, Norway. [2]Princess Margaret Cancer Centre, University Health Network, Toronto, ON, Canada. [3]Institute of Immunology and Infection Research, School of Biological Sciences, University of Edinburgh, Edinburgh, United Kingdom. [4]Department of Health Sciences, Carleton University, Ottawa, Ontario, ON, Canada. ✉e-mail: alberto.d.sanchez@ntnu.no; mennooudhoff@cunet.carleton.ca

crypts are formed after birth, finally, during the weaning stage epithelium fully matures which includes the appearance of Paneth cells. This transition also involves a switch from a non-hierarchical at the fetal stage to a hierarchical stem-cell mediated homeostasis of epithelial turnover in adulthood[9]. Changes in WNT and NOTCH signaling or responses to these pathways are important for this early-life switch[9–11]. In addition, modulators of gene expression, especially those controlling persistent gene expression that can putatively alter cellular state and subsequent intestinal epithelial cell composition, are involved in this change. For example, the transcriptional repressor BLIMP1/PRDM1 is necessary to prevent early enterocyte maturation and Paneth cell differentiation, and thus Prdm1-deficient intestinal epithelium at postnatal day 7 (P7) has accelerated crypt formation and 'adult' epithelium with Paneth cells and matured enterocytes[12,13]. In contrast, the Polycomb Repressive Complex 2 (PRC2) member embryonic ectoderm development (EED) likely regulates the mesoderm-to-epithelial transition that occurs prenatally as it represses these genes in adulthood[14]. In addition, the lysine-specific demethylase 1A (KDM1A/LSD1) is required for optimal postnatal Paneth and goblet cell differentiation and Lsd1-deficient crypts are neonatal-like[15–17]. However, whether LSD1 also controls other aspects of intestinal epithelial maturation, such as enterocyte maturation in villus structures, is still unknown.

The immune cell composition in the intestine also undergoes drastic changes in the fetal to adult timeframe. For example, the adult intestine contains macrophage subsets that originated at the perinatal stage and are long-lived, but also subsets that are blood monocyte-derived and recruited after birth that are short-lived and with a high turnover rate[18]. In addition, both fetal- and postnatal-derived type 2 innate lymphoid cells (ILC2s) contribute to the pool of adult gut-resident ILC2s, which have a slow turnover rate during homeostasis[19]. In contrast, Immunoglobulin A (IgA) is absent at birth, and IgA-secreting plasma cells rapidly appear in the neonatal stage, after which they continue to be important modulators of the microbiota throughout life[20,21]. Gut immune cell homeostasis is in part coordinated by the epithelium. For example, M cells that sample antigen and cover Peyer's patches are important for IgA-producing plasma cells[22], and goblet cells provide antigen during the postnatal stage for regulatory T cell ($T_{reg}$) development[23] and transfer food antigens throughout life for $T_{reg}$ maintenance[24]. Furthermore, tuft cells, which appear after birth[25], are the primary source for IL-25 to control ILC2 numbers both during homeostasis and upon infection[26].

We here provide evidence that epithelial-intrinsic LSD1 is important for the switch from neonatal to adult epithelium. This includes the appearance of Paneth cells in crypts, maturation of other secretory cells such as goblet and enteroendocrine cells, as well as maturation of enterocytes in villus structures. Thus, mice with an intestinal epithelial-intrinsic deletion of Lsd1 (cKO mice) retain a neonatal-like epithelium into adulthood. In addition, deleting Lsd1 later in life reversed adult matured epithelium towards a neonatal-like state. Postnatal epithelial maturation is instrumental for both the commensal microbial composition and immune cell homeostasis. Macrophage subsets, IgA-expressing plasma cells, and ILCs were remarkably dependent on the maturation state of the intestinal epithelium. Importantly, broad-spectrum antibiotics did not interfere with the several epithelial-immune cell interactions we uncovered, suggesting that epithelial maturation has a dominant role in gut immune cell homeostasis.

## Results

### LSD1 controls the differentiation and maturation of intestinal epithelial secretory lineages

We have previously shown that lysine-specific demethylase 1A (KDM1A/LSD1) governs the differentiation of intestinal epithelial secretory cells, including Paneth and goblet cells[15–17]. In our previous studies, we used a Villin-Cre mouse strain that had 10-40% incomplete

recombination[27], which thus led to distinct LSD1+ patches[15,17]. As epithelium-derived factors can have a far-ranging effect in modulating its environment, such as the microbiota and local immune cells, we generated a new mouse line using a different Villin-Cre strain[28]. to completely delete Lsd1 in the intestinal epithelium of both the small intestine (Fig. 1A) and colon (Fig. S1A). Consequently, Villin-Cre+ Lsd1f/f (cKO) mice completely lack Paneth cells throughout the small intestine, as assessed by Lysozyme and Ulex Europaeus Agglutinin 1 (UEA1) staining (and supported by transcriptomic data described below) (Fig. 1A, B). Furthermore, cKO animals have dramatically reduced numbers of MUC2+ goblet cells in both the small intestine (Fig. 1A, C) and the colon (Fig. S1A, B) compared to their wild-type littermates. In addition, goblet cells appeared to have smaller or fewer MUC2+ granules in cKO compared to WT intestinal epithelium, suggesting a lack of maturation (Fig. 1A). To confirm these changes in Paneth and goblet cells and determine how other epithelial cell types are also affected by loss of Lsd1, we performed bulk RNA-seq from crypts and villi to compare intestinal epithelium from wild type (WT) and cKO animals. Like our previous work (ref. 15), we found modest changes in expression levels of intestinal stem-cell markers Lgr5 and Olfm4 or the proliferative marker Mki67, whereas Paneth cell markers Lyz1 and Defa22 were virtually absent in cKO crypts (Fig. 1D and Fig. S1C). Instead, we found that enteroendocrine and tuft cell progenitor markers Neurog3 and Sox4 were much higher expressed in cKO compared to WT crypts (Fig. S1C), which mimics our previous work[15]. In stark contrast, we found markers for mature secretory cells, including tuft (Dclk1, Alox5, Il25), goblet (Muc2, Clca1), and enteroendocrine cells (Chga, Pyy) to be significantly decreased in duodenal villus epithelium (Fig. 1D and Fig. S1C). These transcriptomic changes were not only assessed by canonical markers, but, with the exception of tuft cells, confirmed by performing gene set enrichment analysis (GSEA) of cKO epithelium compared to WT epithelium using cell-specific gene sets from ref. 29 (Fig. 1E). Interestingly, we found fewer DCLK1+ tuft cells in the cKO duodenum (Fig. S1E, F), but this tendency reverts towards the distal portion of the digestive tract (Fig. S1E, G). Nevertheless, we conclude that cKO mice completely lack Paneth cells and that their goblet cells are strongly reduced in number and insufficient in maturation. In addition, although cKO crypts are enriched for enteroendocrine and tuft cell progenitor marker gene expression, cKO cells fail to formally mature and lack mature secretory cell markers in the villus as assessed by bulk RNA-seq.

### LSD1 controls enterocyte maturation, independent of BMP signaling

Enterocytes differentiate along the crypt-villus axis, with the least-differentiated enterocytes located at the villus-base and the most-differentiated enterocytes near the villus-tip, a maturation process mediated by a BMP gradient[30–32]. To assess enterocyte maturation, we leveraged ref. 30 villus enterocyte-specific zonation gene sets and compared our cKO and WT villus transcriptomes to these spatially-restricted maturation signatures (see C1–C5 in Fig. 1F). We found an enrichment of "clusters 1–2" and overall lower levels of genes associated with the more mature "clusters 3–5" in cKO complete villus epithelium (Fig. 1F). In support, by GSEA, we find enrichment of an "enterocyte-progenitor" gene set but repression of a "mature-enterocyte" gene set when comparing cKO and WT villus by bulk RNA-seq (Fig. 1E). This suggests that cKO villi are filled with immature enterocytes that would normally only reside at the villus-base. To define this in more detail, we performed in situ hybridization (ISH) of key cellular marker genes (Lyz1, Muc2, Reg1, Nt5e, Ada). Indeed, we find that secretory lineage markers (Lyz1 and Muc2), as well as villus-tip enterocyte markers Nt5e and Ada are absent or reduced in cKO epithelium (Fig. 1G and Fig. S1D). In addition, we observed a change in distribution for villus-base marker Reg1, going from villus-base-restricted in WT animals to throughout the duodenal villus structure in cKO littermate

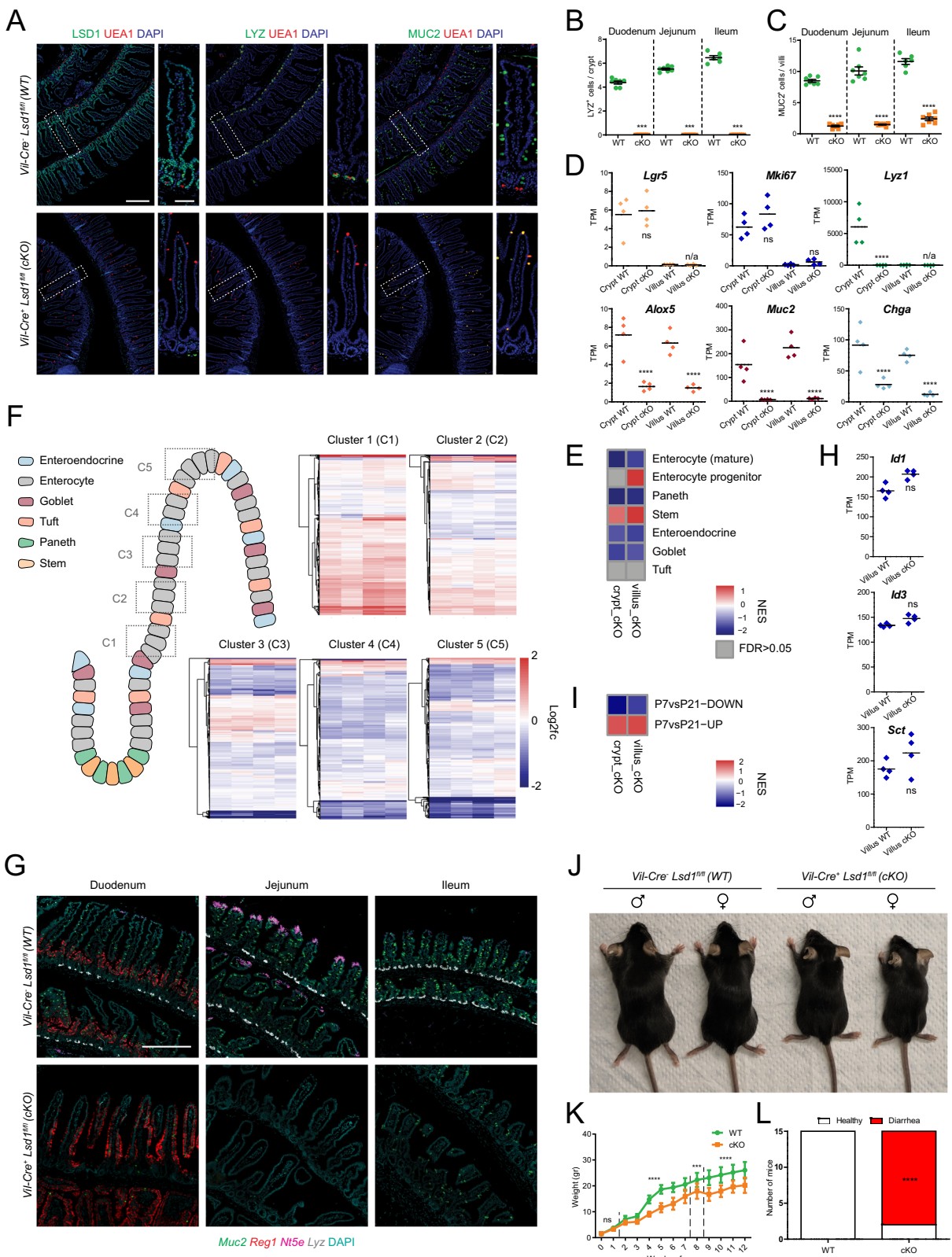

animals (Fig. 1G). These data may suggest that LSD1 is required for the ability of intestinal epithelium to respond to BMP ligands to mature enterocytes upon the villus axis. However, canonical BMP target genes *Id1* and *Id3*, as well as enteroendocrine-specific BMP target gene *Sct*[33], are equal to or even higher expressed in cKO compared to WT villus epithelium (Fig. 1H). Therefore, instead, our data suggests that cKO villus epithelium retains a neonatal-like state into adulthood, rather

than the inability to respond to BMP signals. Indeed, by using gene sets originating from transcriptomics comparing intestinal epithelium from postnatal day 7 (P7) *vs.* P21 (from ref. 15), we find that genes highly expressed at the P7 stage are enriched in cKO epithelium, whereas P21-upregulated genes are repressed (Fig. 1I).

In our previous study, we did not find overt abnormalities in *Villin*-Cre *Lsd1*^f/f mice (with incomplete deletion), when unchallenged, they

**Fig. 1 | LSD1 is required for the postnatal maturation of the intestinal epithelium. A** Immunofluorescence staining of LSD1, Lysozyme (LYZ), MUC2, and UEA1 of paraffin-embedded duodenal tissue from WT and cKO 2-month-old littermates. Scale bar: 200 μm, inset scale bar: 50 μm. **B, C** Quantification of small-intestinal Paneth (LYZ+) and goblet (MUC2+) cells in WT and cKO mice (derived from images such as shown in Fig. 1A). Data were presented as mean ± SEM; *n* = 7 mice/genotype from two independent experiments (except 6WT/Ileum), (Two-tailed Mann−Whitney non-parametric test for LYZ+ data, Two-tailed unpaired *t*-test for MUC2+ data). **D** Bulk RNA-seq of crypt and villus fractions derived from WT and cKO 2-month-old mice. Individual graphs show transcripts per million (TPM). Data were presented as mean and individual data points; *n* = 4 mice/genotype, (Differential expression analyzed using DESeq2's negative binomial generalized linear model, with Benjamini−Hochberg adjusted *p* values). **E** Heatmap showing normalized enrichment scores (NES) from gene set enrichment analysis (GSEA) of cell type specific gene sets from ref. [29] (see Supplementary Data 1) cross-referenced against Fig. 1D bulk RNA-Seq. Crypt cKO is compared to crypt WT, and villus cKO is compared to villus WT. **F** Schematic representation of enterocyte-specific gene expression zonation clusters[30] along the intestinal villus axis (C1−C5). Heatmaps depict a comparison of our villus-derived bulk RNA-seq data against said clusters. Each row is a gene from the gene set, and each column represents a biological replicate of villus cKO compared to the mean value of villus WT. Log2fc is capped at −2 and 2. **G** Representative fluorescence in situ hybridization of cellular markers for goblet (*Muc2*, green), Paneth (*Lyz*, white), villus-base (*Reg1*, red), and villus-tip (*Nt5e*, magenta) enterocytes. Scale bar: 250 μm; *n* = 5 mice/genotype from two independent experiments. **H** BMP target genes. TPMs derived from Fig. 1D bulk RNA-Seq. Data are presented as mean and individual data points; *n* = 4 mice/genotype, (Differential expression analyzed using DESeq2's negative binomial generalized linear model, with Benjamini−Hochberg adjusted *p* values). **I** Heatmap of NES from GSEA of mouse intestinal epithelium gene expression signatures derived from postnatal day 7 (P7) vs P21 comparisons[15] (see Supplementary Data 1) cross-referenced against Fig. 1D bulk RNA-Seq. Crypt cKO is compared to crypt WT, and villus cKO is compared to villus WT. **J** Representative picture of male/female 8-week-old mice. **K** Gender aggregated weights are presented as mean ± SEM; *n* = 12 mice, (six male and six female)/genotype/timepoint from four independent experiments, (Individual timepoints assessed by two-tailed unpaired *t*-test, timepoints with the same significance level are shown aggregated and separated by discontinuous lines). **L** Diarrhea incidence evaluation by stool consistency. *n* = 15 mice/genotype from four independent experiments, (Contingency analysis by two-sided Fisher's exact test). Cell nuclei in all imaging are counterstained with DAPI. Source data and exact *p* values are provided as a Source Data file.

lived up to at least a year without issues[15]. In contrast, our newly generated cKO animals failed to normally gain weight after weaning (Fig. 1J, K). In addition, within the first 12 weeks of life, 13 out of 15 cKO animals developed diarrhea (Fig. 1L). The timing by which the lack of gaining weight started (weeks 2−4), suggests that cKO animals start to have issues upon switching from milk to solid food, a time window particularly important in intestinal maturation[1]. We envision many factors could cause the lack of weight gain, including a lack of properly functioning enterocytes for absorption, impaired enteroendocrine cell function, or a putative altered microbiota. In addition, diarrhea can be a sign of inflammation caused by barrier disruption. Therefore, we assessed the epithelial barrier by E-cadherin (adherens junctions) staining and found no abnormalities between WT and cKO tissues (Fig. S1H). This suggests that although the epithelium is strongly altered, there is no sign of barrier disruption. Therefore, we conclude that cKO animals retain a neonatal-like intestinal epithelial cell state into adulthood, which includes the absence of Paneth cells, reduced numbers and maturation of other secretory cells, and immature enterocytes. Yet, a clear distinction between crypt and villus, structures that are formed in the first week after birth, remains.

## Mature intestinal epithelium guides post-weaning microbiota development

Next, we wanted to test whether maturation of intestinal epithelium affects microbiota establishment and development. We therefore performed 16S ribosomal RNA sequencing of samples isolated from stool and compared WT and cKO litter and cage mate animals at neonatal (P14), before switching from milk to solid food, and adult life stages (6 weeks). By principal coordinate analysis (PCoA) we found the microbiome was indistinguishable at P14, but substantially different at 6 weeks (Fig. 2A). Similarly, the microbiota diversity changes (Shannon index) were non-significant when comparing P14 samples but highly significant when comparing adult WT and cKO littermates (Fig. S2A). Of note, microbiome in cKO does not retain a neonatal state but develop towards a different composition with notably lesser percentages of *Muribaculaceae* and *Rikenellaceae* and increased percentages of *Bacteroidaceae* and *Sutterellaceae* (Fig. 2B). Notably, the observed shifts in the *Bacteroidaceae:Muribaculaceae* ratio in cKO as well as the severity of the changes in the microbiota could be a consequence of diarrhea. Indeed, similar shifts were found in mice that were given a laxative causing mild osmotic diarrhea[34]. Osmotic diarrhea can develop due to malabsorption, and absorption is likely to be impaired in cKO mice. In particular, genes associated with absorption of peptides (*Slc15a1*), fatty acids (*Cd36*), and lipids such as cholesterol (*Apob, Npc1l1*) are significantly reduced in cKO villi (Fig. S2B).

Maturation of the intestinal epithelium is important for gut microbial composition establishment (Fig. 2A, B). Next, we wished to test whether matured epithelium is also important for the maintenance of the microbial composition. We, therefore, generated *Villin*-CreERT2 *Lsd1*f/f (icKO) mice in which we completely deleted *Lsd1* from small intestinal and colon epithelium in adulthood by tamoxifen administration (Fig. 2C, D and Fig. S2C). We found that as early as 2 days after five consecutive tamoxifen administration by oral gavage (wk1 timepoint), the intestinal epithelium is completely transformed. We observed a stark reduction in goblet cells, a loss of the mature-enterocyte villus-tip marker *Nt5e*, and an expansion of the immature-enterocyte villus-base marker *Reg1* that now covers the whole villus (Fig. 2D). Similarly, in the colon, we see a rapid reduction of *Muc2*+ goblet cells, especially in the upper half of the crypt where fully matured goblet cells reside (Fig. 2D). Although most intestinal epithelium turns over within a week, Paneth cells have an estimated lifetime of 4−6 weeks, and indeed only 4 weeks after *Lsd1* deletion did we observe a complete loss of *Lyz1*+ Paneth cells (Fig. 2D). Thus, LSD1 is required for Paneth cell differentiation and not their maintenance, unlike other modulators such as ATOH1 where deletion leads to Paneth cell depletion within a week[35]. We hereby establish a model to reverse matured intestinal epithelium towards a status that recapitulates features of neonatal/developing epithelium. After LSD1 deletion, icKO animals failed to gain weight at the same rate as their WT counterparts, but only 4 out of 17 animals developed diarrhea (Fig. 2E, F), suggesting that the lack of weight gain in cKO mice is due to a lack of intestinal epithelial maturation and not a consequence of diarrhea.

We collected stool from icKO and WT littermates before and every week after tamoxifen treatment for 6 weeks and performed 16S ribosomal RNA sequencing. Although we recapitulated similar population shifts in icKO mice compared to those in cKO mice, especially from week 4 onwards (e.g., decreased *Muribaculaceae* and *Rikenellaceae* and increased percentages of *Bacteroidaceae* and *Sutterellaceae*), they were much more subtle in icKO mice (Fig. 2B, G and Fig. S2D). Before tamoxifen administration, WT and icKO gut microbiomes were indistinguishable and gradual changes linked to the loss of specific intestinal epithelial cells could be observed at different timepoints (Fig. 2H). For example, around the 3 to 4 weeks post-tamoxifen timepoint, mucus-associated bacteria, such as *Sutterellaceae* and *Bifidobacteriaceae*[36,37], suffered a sharp decrease coinciding with the loss of most goblet and Paneth cells across the intestine (Fig. 2D and Fig. S2C, D). In addition, around the same timepoint, opportunistic bacteria (from *Acidaminococcaceae* to *Fusobacteriaceae*) began occupying the ecological niche that was vacated by other predominant bacterial populations (Fig. S2D). Finally, comparing gut microbiomes of WT, cKO, and icKO

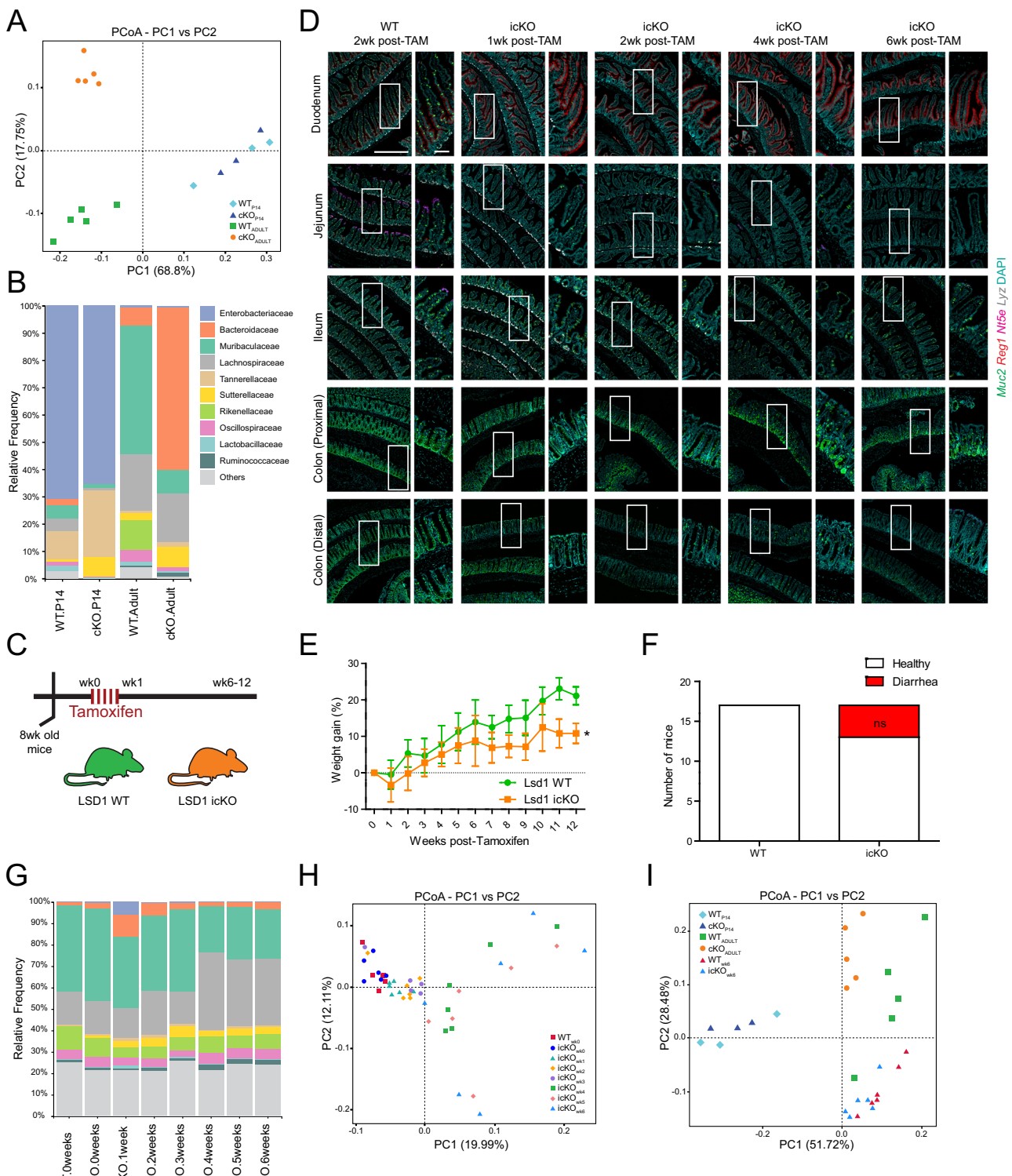

**Fig. 2 | Mature intestinal epithelium defines and maintains microbial composition. A** Principal coordinate analysis (PCoA) of stool-derived bacterial 16S rRNA was obtained from adult (2-month-old) and P14 mice (WT and cKO); $n = 5$ adult mice/genotype and $n = 3$ P14 mice/genotype. **B** Average relative frequency of bacterial families found in Fig. 2A stool samples. **C** Tamoxifen treatment scheme used to induce complete deletion of *Lsd1* across the intestinal epithelium. wk0 = first day of tamoxifen oral gavage, wk1 = 7 days after first tamoxifen oral gavage. Stool and intestinal tissue samples were collected at wk1, wk2, wk4, and wk6 timepoints. **D** Representative images of fluorescence in situ hybridization of key cellular markers for goblet (*Muc2*, green), Paneth (*Lyz*, white), villus-base (*Reg1*, red), and villus-tip (*Nt5e*, magenta) enterocytes. Scale bar: 500 μm, inset scale bar:

100 μm; $n = 5$ mice/genotype (Duo, Jej, and Ile) and $n = 3$ mice/genotype (Colon) from two independent experiments. **E** Gender aggregated weight assessment of WT and icKO mice post-tamoxifen administration. Weights are presented as mean ± SEM; $n = 4$ mice, (two male and two female)/genotype/timepoint (Two-way repeated measures ANOVA). **F** Diarrhea incidence evaluation by stool consistency. $n = 17$ mice/genotype from four independent experiments (Contingency analysis by two-sided Fisher's exact test). **G** Average relative frequency of bacterial families in stool derived from icKO mice before (icKO.0weeks) and after (icKO.1–6 weeks) tamoxifen administration; $n = 6$ mice/genotype/timepoint. **H** PCoA representation of Fig. 2G samples. **I** Combined the PCoA representation of Fig. 2A, H samples. Source data and exact $p$ values are provided as a Source Data file.

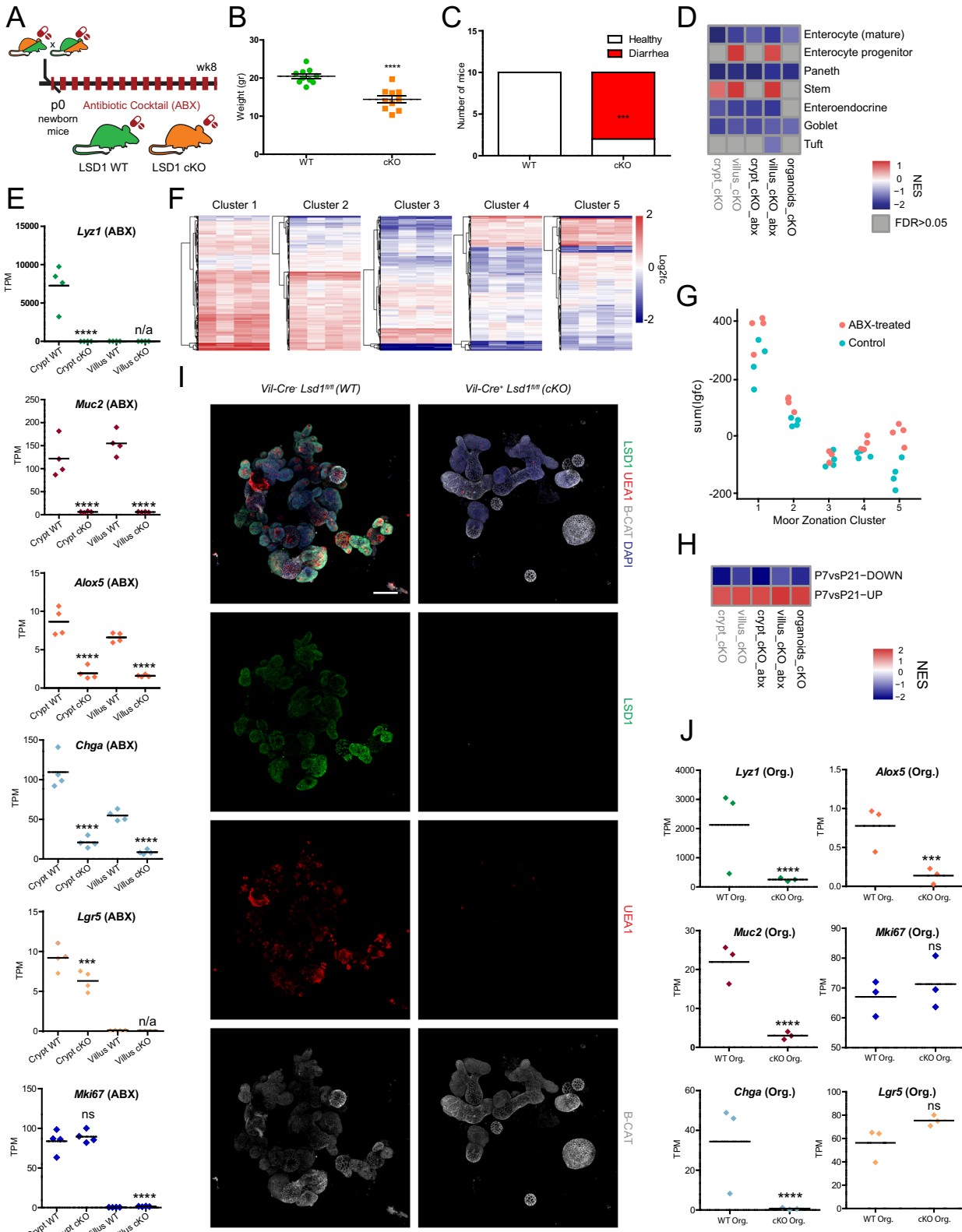

mice we confirm that cKO animals have stronger microbiota alterations than icKO mice when compared to respective WT littermates (Fig. 2I), which may be related to the diarrhea incidence in cKO mice.

## LSD1-driven epithelial maturation is independent of the microbiota

The microbiota is an established modulator of intestinal epithelial development. For example, microbiota-derived metabolites can alter intestinal epithelial biology[38,39], and antibiotics-treated animals lack full maturation of intestinal epithelium[40,41]. To test whether the altered microbiota is, in part, responsible for the effect of LSD1 on the intestinal epithelium, we treated pregnant mothers and their offspring with a cocktail of antibiotics (ABX, see M&Ms Fig. 3A), ensuring thorough depletion of the microbiota, as verified by 16S rRNA qPCR (Fig. S3A). Like conventionally bred animals (not treated with ABX, see M&Ms), we found that cKO animals on ABX did not gain weight

**Fig. 3 | LSD1-driven epithelial maturation is independent from the microbiota.**
**A** Antibiotic cocktail (ABX) treatment scheme. Mating pairs, pregnant mothers, and offspring were treated with a cocktail of five antibiotics (see M&Ms) during their entire lifetime. **B** Gender aggregated weights are presented as mean ± SEM; $n = 10$ mice (five male and five female)/genotype from three independent experiments (Two-tailed unpaired $t$-test). **C** Diarrhea incidence evaluation by stool consistency. $n = 10$ mice/genotype from three independent experiments (Contingency analysis by two-sided Fisher's exact test). **D** Heatmap showing normalized enrichment scores (NES) from gene set enrichment analysis (GSEA) of cell type specific gene sets[29] (see Supplementary Data 1) cross-referenced against bulk RNA-Seq of crypts/villi fractions collected from adult WT (ABX) and cKO (ABX) mice. For each comparison, tissue and antibiotic status were matched, i.e., villus LSD1 cKO ABX is compared to villus WT ABX. Bulk RNA-Seq of organoids obtained from untreated WT and cKO mice is included in the comparison. Samples not treated with ABX from Fig.1E are included as references (gray text). **E** Bulk RNA-seq of crypt and villus fractions derived from WT (ABX) and cKO (ABX) 2-month-old mice. Individual graphs show transcripts per million (TPM). Data were presented as mean and individual data points; $n = 4$ mice/genotype, (Differential expression analyzed using DESeq2's negative binomial generalized linear model, with Benjamini–Hochberg adjusted $p$ values). **F** Heatmaps depict a comparison of our villus-derived (ABX) bulk RNA-seq data against enterocyte zonation clusters (see Fig. 1F). Each row is a gene from the gene set, and each column represents a biological replicate of LSD1 cKO villus ABX-treated compared to the mean value of WT villus ABX-treated. Log2fc is capped at −2 and 2. **G** Dot plot of the summed log2(fold change) in genes from the different zonation clusters from the intestinal epithelium. Each dot represents the sum of a column in Figs. 1F, 3F. **H** Heatmap NES from GSEA of mouse intestinal epithelium gene expression signatures derived from postnatal day 7 (P7) vs P21 comparisons[15] (see Supplementary Data 1) cross-referenced against same bulk RNA-Seq data described in Fig. 3D. Untreated samples from Fig. 1I are included as reference (gray text). **I** Organoids from untreated WT and cKO mice were cultured for 3 days. LSD1 (green), β-catenin (white), and goblet/Paneth cell (UEA1+) presence was determined by immunofluorescence; $n = 3$ mice/genotype. Scale bar: 100 μm. **J** Bulk RNA-seq of organoids derived from untreated WT and cKO mice. Individual graphs show transcripts per million (TPM). Data were presented as mean and individual data points; $n = 3$ mice/genotype (Differential expression analyzed using DESeq2's negative binomial generalized linear model, with Benjamini–Hochberg adjusted $p$ values). Source data and exact $p$ values are provided as a Source Data file.

properly and most developed diarrhea (Fig. 3B, C), suggesting that dysbiosis (or changes in the microbiome) did not cause these phenotypes. Indeed, we observe the same lack of expression of genes associated with the absorption of peptides (*Slc15a1*), fatty acids (*Cd36*), and lipids such as cholesterol (*Apob, Npc1l1*) (Fig. S3B).

In addition, by comparing transcriptomes of WT and cKO crypts and villi (derived from ABX mice), we observed a remarkably similar pattern as found in conventionally raised animals: We find complete loss of Paneth cell markers and reduced expression of all other epithelial secretory lineages (Fig. 3D, E and Fig. S3B). Furthermore, we find an enrichment of genes expressed at the villus-base (clusters 1–2) and reduced expression of genes associated with the villus-tip (clusters 3–5) (Fig. 3F, G). Using cell type specific gene sets, we find enrichment of the progenitor enterocyte gene set and repression of a mature-enterocyte gene set in ABX-treated cKO animals (Fig. 3D), as well as enrichment of P7 and repression of P21 signatures (Fig. 3H). Of note, we did observe some differences between conventionally raised and ABX treatment. For example, the stem-cell-associated gene *Lgr5* is modestly reduced in ABX-treated cKO vs. WT crypts, whereas it is unchanged in untreated animals (Fig. 1D, 3E). In addition, we found upregulation of a number of genes in cKO villus epithelium among the regional gene set clusters 4 and 5 upon ABX treatment (Fig. 3F, G), whereas almost all those genes were downregulated in conventional animals (Fig. 1F).

Furthermore, we generated organoids from untreated WT and cKO animals and found that also in vitro, thus isolated from external factors such as microbiota or immune cells, LSD1-deficient organoids (cKO Orgs.) lacked UEA1-positive Paneth and goblet cells (Fig. 3I). Although organoids are considered somewhat immature versions of the adult intestinal epithelium, we still observed reduced expression of the mature-enterocyte gene set and enrichment of the overall P7 *vs.* P21 gene sets comparing cKO to WT organoids (Fig. 3D, H) as well as reduced expression of mature cell markers associated with Paneth, goblet, enteroendocrine, and tuft cells (Fig. 3J and Fig. S3C). Finally, in order to control for potential microbial imprinting of stem cells, we induced *Lsd1* recombination in vitro by administering tamoxifen to organoids from icKO mice that had not received tamoxifen previously. Here, *Lsd1* deletion resulted in gene expression profiles that included loss or reduction of matured intestinal epithelial cell markers (*Lyz1 & Defa22* for Paneth cells, *Muc2 & Clca1* for goblet cells, *Chga* for enteroendocrine cells and *Ada & Nt5e* for mature enterocytes, Fig. S3D). Overall, we observed the characteristic absence of UEA1-positive cells (Fig. S3E, G) and granular Paneth cells (Fig. S3F, H) in tamoxifen-treated icKO organoids.

Together, this body of evidence shows that LSD1 directs the intestinal epithelial maturation process intrinsically, independently of

external factors such as the microbiota (ABX-derived data) or the tissue niche (organoid-derived data).

## LSD1-mediated intestinal epithelial maturation does not control systemic immune cell imbalance in the spleen or mesenteric lymph nodes

Both the intestinal epithelium and gut microbiota are dramatically different when comparing adult cKO and WT mice (Figs. 1–3). The intestinal epithelium, as well as the microbiota, are crucial mediators of intestinal and even systemic inflammation. We were, therefore, surprised that we did not observe any signs of tissue inflammation in cKO animals by gross pathology. For example, splenomegaly and enlarged mesenteric lymph nodes (mLN) are associated with a persistent inflammatory response in the gut. However, upon isolation of spleens and mLNs from 6-8-week-old cKO mice and their WT littermates, we did not observe differences in spleen size (Fig. 4A, B). In addition, we did not find differences in total cell numbers isolated from mLNs when comparing those from cKO and WT mice (Fig. 4C). Moreover, no overt changes in the immune cell composition (T cells, B cells and antigen-presenting cells) were observed in either spleen (Fig. S4A) or mLN-derived CD45+ cells (Fig. S4B), as assessed by flow cytometry (see gating hierarchy and strategy, Fig. S4C–E). This included CD4$^+$ CD25$^+$ regulatory T (T$_{reg}$) cell numbers (Fig. S4A, B).

## Intestinal epithelial maturation directs immune cell populations in the lamina propria

To test the role of the maturation of intestinal epithelium in the composition of local gut immune cells, we performed single-cell RNA-seq (scRNA-seq) on sorted lamina propria CD45+ cells. We were able to distinguish most immune cell types expected to be present (Fig. 4D and Fig. S4F). As the microbiota is also known to affect local gut immune cell populations and we wanted to dissect the role of intestinal epithelial maturation in the process, we not only performed this experiment in naive cKO and WT animals but also in WT and cKO mice that received ABX (Figs. 3A, 4D). Besides a decrease in *Il10*-producing macrophages, and corroborating our finding that cKO mice do not develop overt inflammation, we did not observe drastic changes in the expression of inflammation-associated cytokines (*Ifng, Il1b, Tnf, Il22,* and *Il17a*) in cells derived from cKO animals compared to those from WT animals (Fig. S4G). Next, we grouped cells either by genotype (i.e., WT *vs.* cKO, thus pooling cells from conventionally bred and ABX-treated mice) or by treatment (i.e., conventionally bred *vs.* ABX, thus pooling cells from both WT and cKO mice) (Fig. 4E, F). We found that the putative cellular state, as inferred from a change in UMAP clustering, of dendritic cells (DCs), plasmacytoid DCs (pDCs), and IgA-

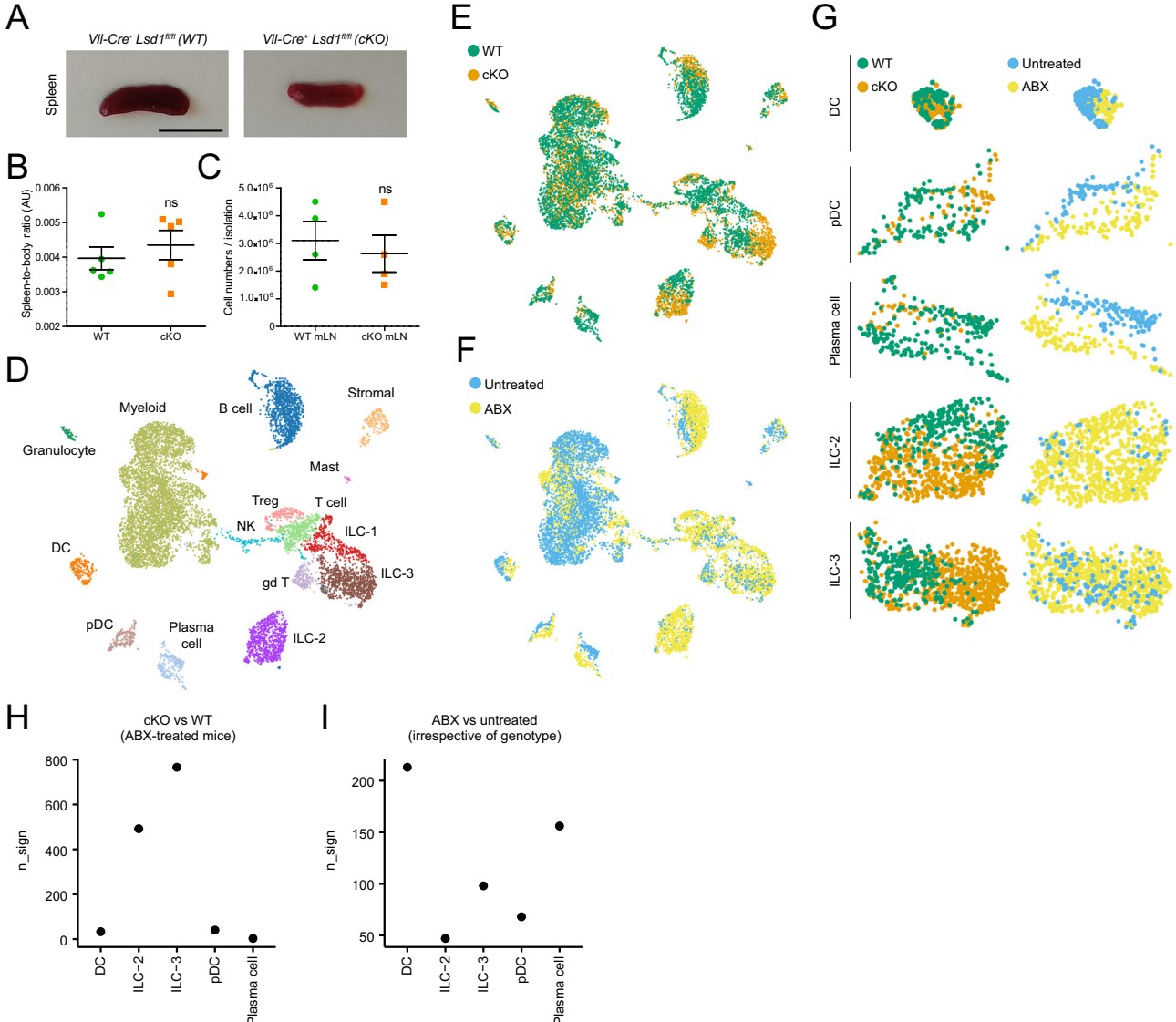

**Fig. 4 | LSD1-mediated intestinal epithelial maturation does not control systemic immune cell imbalance in the spleen or mesenteric lymph nodes but directs local immune cell populations. A** Representative picture of spleens derived from 2.5-month-old mice. Scale bar: 1 cm; *n* = 5 mice/genotype. **B** Spleen-to-body weight ratio (arbitrary units, AU). Data were presented as mean ± SEM; *n* = 5 mice/genotype from four independent experiments (Two-tailed Mann–Whitney non-parametric test). **C** Number of cells derived from mesenteric lymph nodes (mLN) single-cell suspension isolation. Data were presented as mean ± SEM; *n* = 4 mice/genotype from four independent experiments (Two-tailed unpaired *t*-test). **D** UMAP (uniform manifold approximation and projection) plot shows cell types as determined by scRNA-seq from FACS-sorted CD45+ cells derived from the small intestinal lamina propria of ABX and untreated WT and cKO mice. Each immune cell type class is represented as a cluster of points in a unique color; *n* = 1 mouse/ genotype/condition. **E** Overlay of genotype (WT or cKO) class over UMAP clustering of all experimental conditions described in Fig.4D. **F** Overlay of treatment (untreated or ABX) class over UMAP clustering of all experimental conditions described in Fig.4D. **G** Detailed DC, pDC, Plasma cell, ILC2, and ILC3 UMAP clusters derived from Fig. 4E, F. **H** Number of differentially expressed genes (n_sign, *p* < 0.05) when comparing genotype variable (cKO vs WT) per immune cell type in ABX-treated mice; *n* = 1 mouse/genotype (Two-sided Wilcoxon rank test). **I** Number of differentially expressed genes (n_sign, *p* < 0.05) when comparing treatment variable (ABX vs untreated, aggregating WT with cKO untreated and WT ABX with cKO ABX samples) per immune cell type; *n* = 1 mouse/genotype/condition (Two-sided Wilcoxon rank test). Source data and exact *p* values are provided as a Source Data file.

expressing plasma cells is determined by treatment (i.e., conventional vs. ABX), hence by the presence or absence of microbiota (Fig. 4G). In contrast, type 2 and 3 innate lymphoid cells (ILC2s and ILC3s), offer a random distribution when grouping them according to treatment, but display a distinct pattern by genotype, and thus by the maturation state of intestinal epithelium (Fig. 4G). In support, we found higher numbers of differentially expressed genes (n_sign) in ILC2s and ILC3s derived from WT and cKO ABX-treated mice as opposed to DCs, pDCs, and plasma cells (Fig. 4H), while the opposite (except for pDCs) was observed when aggregating WT plus cKO samples and comparing conventional vs. ABX-treated mice (Fig. 4I).

### LSD1-mediated intestinal epithelial maturation controls intestinal plasma cell numbers and secretory IgA levels

Plasma cells secrete large amounts of IgA to modulate the microbiota and thereby maintain intestinal homeostasis[21]. In our scRNA-seq, we found that there were reduced numbers of plasma cells detected in CD45+ sorted cells from cKO mice compared to WT mice, irrespective of ABX treatment (Fig. 5A). Next, we stained small intestinal sections for IgA and confirmed lower number of IgA+ cells in small intestines of cKO compared to WT (Fig. 5B). Similarly, IgA levels in small intestinal content were significantly lower in those derived from cKO animals compared to WT animals, irrespective of treatment (Fig. 5C). Overall, ABX

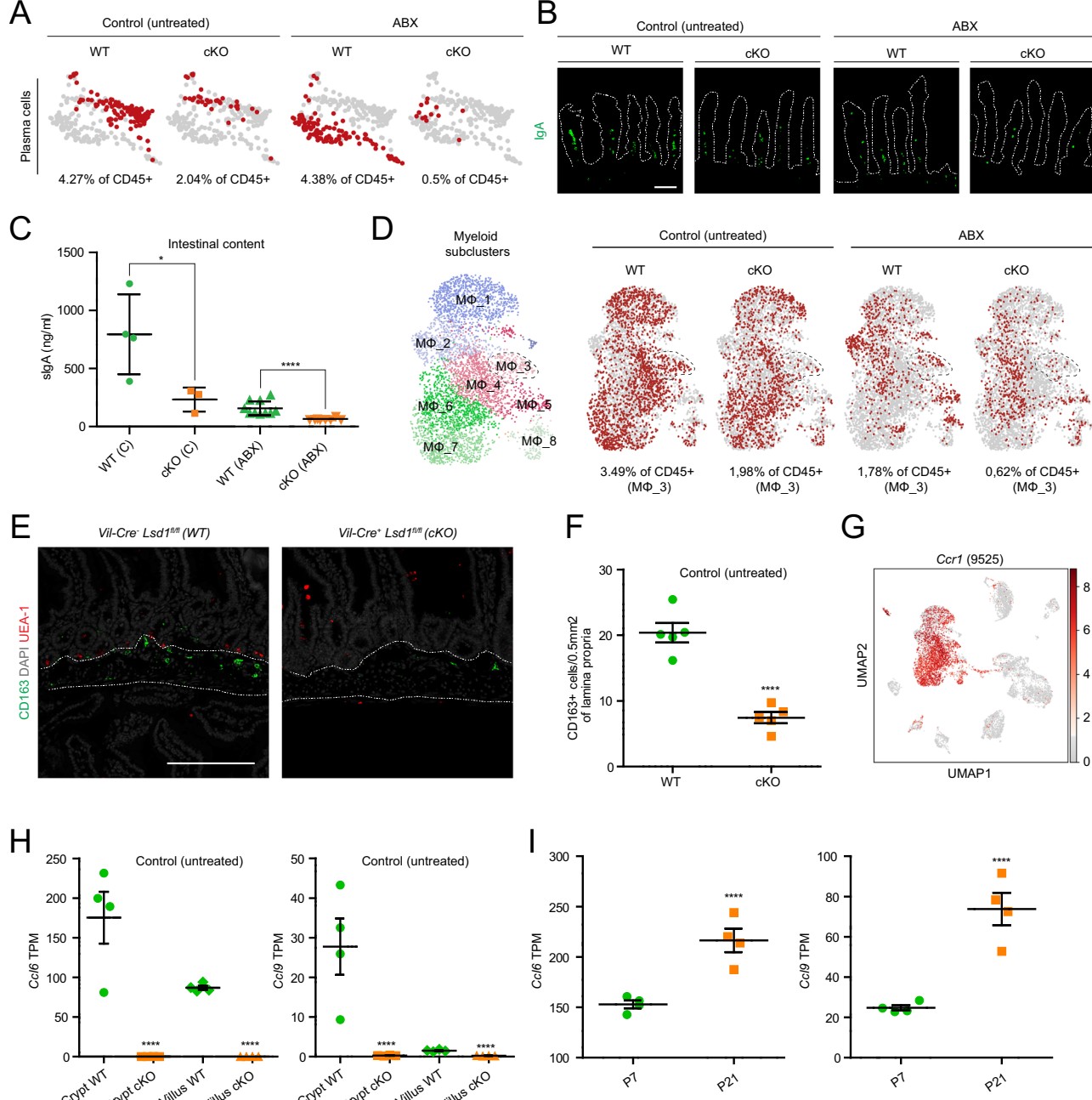

**Fig. 5 | LSD1-mediated intestinal epithelial maturation controls intestinal plasma cell and macrophage homeostasis. A** UMAP clustering of lamina propria plasma cells across all experimental conditions. Red dots correspond to the number of plasma cells detected under each condition. Gray dots represent all detected plasma cells across conditions. **B** Immunofluorescence staining of paraffin-embedded mouse duodenal tissue depicting IgA-producing plasma cells in green. Scale bar: 100 μm. Villus structure is delimited by a discontinuous white line; $n = 5$ mice/genotype/condition from two independent experiments. **C** Secreted IgA (sIgA) protein concentration as determined by ELISA on small intestinal content of 2-month-old WT and cKO mice (untreated and ABX-treated). Data were presented as mean ± SEM; $n = 4$WT/3cKO mice (untreated) and $n = 10$ mice/genotype (ABX) from three independent experiments (Two-tailed Mann–Whitney non-parametric test). **D** UMAP subclustering of lamina propria myeloid population in eight different populations (MΦ_1-8) across all experimental conditions. MΦ_3 is highlighted across conditions by a discontinuous line (% of CD45 corresponds to this subcluster). **E** Paraffin section of mouse duodenum submucosal region. CD163+ macrophages (green) are detected by immunofluorescence, UEA1+ cells (red) correspond to Paneth and/or goblet cells. Nuclei counterstained with DAPI (gray).

Scale bar: 200 μm; $n = 5$ mice/genotype from two independent experiments (untreated). **F** CD163+ cell density in the intestinal submucosal region. Data were presented as mean ± SEM; $n = 5$ mice/genotype (untreated) from two independent experiments (untreated) (Two-tailed unpaired $t$-test). **G** UMAP of lamina propria CD45+-derived cells showing *Ccr1* gene expression across all experimental conditions. The number in between parentheses represents the number of sequenced cells that passed quality control (9525) and corresponds to all four conditions merged under one UMAP. Gray to red heatmap scale shows log(CPM+1). **H** Bulk RNA-seq of crypt and villus fractions derived from WT and cKO untreated 2-month-old mice. Individual graphs show transcripts per million (TPM). Data were presented as mean ± SEM; $n = 4$ mice/genotype, (Differential expression analyzed using DESeq2's negative binomial generalized linear model, with Benjamini–Hochberg adjusted $p$ values). **I** Bulk RNA-seq of intestinal epithelium derived from WT mice at postnatal day 7 (P7) vs P21[15]. Individual graphs show transcripts per million (TPM). Data were presented as mean ± SEM; $n = 4$ mice/timepoint, (Differential expression analyzed using DESeq2's negative binomial generalized linear model, with Benjamini–Hochberg adjusted $p$ values). Source data and exact $p$ values are provided as a Source Data file.

treatment led to reduced numbers of IgA+ cells and lower levels of IgA in intestinal content (Fig. 5B, C). In part, plasma cells originate in Peyer's patches where specialized epithelial M cells transfer antigen from the lumen, and mice lacking M cells have reduced IgA levels in a similar manner to what is observed in our cKO animals[22]. In addition, goblet cells can passage antigen, especially early in life[23,24], and goblet cells are clearly affected in cKO animals (Figs. 1–3). Unfortunately, we were not able to stain for M cells nor did we assess Peyer's patch epithelium by RNA-seq. In future work, we hope to address M cell differentiation, antigen uptake, and plasma cell development in more detail.

### LSD1-mediated intestinal epithelial maturation is required for a putative CCL6/CCL9-CCR1 axis in macrophage establishment or recruitment

In our scRNA-seq data, we identified one large myeloid population (Fig. 4D) consisting of eight myeloid subclusters (MΦ_1-8) (Fig. 5D). Upon assessing individual panels, we found that ABX treatment drastically reduces the percentage of total myeloid cells, as is commonly found by others[42,43]. In contrast, myeloid population 2 (MΦ_2) appeared upon ABX treatment (Fig. 5D). We were unable to assess whether this is due to a shift in an existing population, or an appearance and recruitment of a separate one. Next, we focused on changes in cKO compared to WT-derived cells. In addition to the overall lower percentage of myeloid cells in cKO, we found that myeloid population 3 (MΦ_3) was nearly absent (Fig. 5D). This population co-expressed markers such as *F13a1* and *Cd163* (Fig. S5A), and was recently identified to mark a specific small-intestinal macrophage population residing underneath intestinal crypts[44]. Indeed, upon staining with an anti-CD163 antibody, we confirmed this specific location and reduced numbers in the small intestine of cKO mice compared to those from WT mice (Fig. 5E, F).

Chemokine receptors are important factors in macrophage gut homeostasis, indeed the CCL2-CCR2 axis is crucial for monocyte recruitment from blood which subsequently differentiate into macrophages in the tissue[45,46]. Within intestinal tissue, similarly to other tissues, CCR2$^+$ and CCR2$^−$ MΦ populations co-exist, identifying recently differentiated MΦ and long-lived MΦ, respectively[47]. Our population of interest, MΦ_3, included a mixture of *Ccr2*-expressing and non-expressing cells (Fig. S5B). Notably, however, most myeloid cells expressed *Ccr1*, including MΦ_3 (Fig. 5G and Fig. S5B). In search of changes in corresponding chemokine receptor-ligand expression of the intestinal epithelium, we found that *Ccl6* and *Ccl9*, which encode CCR1 receptor ligands CCL6 and CCL9, respectively, were nearly completely absent in cKO epithelium, irrespective of ABX treatment (Fig. 5H and Fig. S5C). In line with the submucosal localization of CD163$^+$ macrophages, both *Ccl6* and *Ccl9* show higher expression in crypt fractions compared to villus fractions (Fig. 5H and Fig. S5C). Both these chemokines are expressed in the goblet and a small subset of Paneth cells as deducted from published scRNA-seq data from ref. 29 (Fig. S5D). In addition, CCL6 has been previously found to be enriched at the bottom of crypts of both the colon and small intestine[48]. Importantly, both *Ccl6* and *Ccl9* are markers that are drastically induced at P21 compared to P7 (Fig. 5I), and thus part of the early-life intestinal maturation that is governed by LSD1 and may contribute to the establishment of CD163$^+$ macrophages within this niche.

### LSD1-mediated intestinal epithelial maturation is required for the establishment and maintenance of ILC2s and ILC3s in a microbiota-independent manner *via* a putative CXCL16-CXCR6 axis

We found that the cellular state, as inferred from scRNA-seq clustering and differential gene expression, of small intestinal ILC2s and ILC3s is dependent on genotype (Fig. 4G, H). Overall, gut-resident ILCs arrive, expand, and/or mature early in life and are tuned by the presence of the microbiota[19,49]. Therefore, our next step was to assess ILC2 and

ILC3 numbers in adult tissue. We found higher relative ILC2 and ILC3, but not ILC1 numbers in the CD45+ population by scRNA-seq when comparing WT with cKO-derived cells (Fig. S6A). Next, we quantified cells using fluorescent immunostaining of small intestinal sections. ILC2s were detected and quantified using CD3- RORγt- GATA3+ as markers, and ILC3s were defined as CD3- RORγt+ entities. In support of the scRNA-seq data, both these cell types were detected in increased quantities in cKO mice compared to WT mice, both in the presence (untreated controls) and absence (ABX) of microbiota (Fig. 6A–C and Fig. S6B–D). Finally, increased numbers of Lin- CD11b- CD127+ NKp46+ (CD335) cells, a gating that largely overlaps with a predominant subset of lamina propria ILC3s[50,51], was also confirmed by surface staining using flow cytometry (Fig. 6D).

The intestinal epithelium is known to be able to locally control ILCs by expressing specific factors. For example, ILC2s respond to epithelial-derived IL-25, IL-33, and TSLP[52–55], while IL-15 and IL-18[56,57] can promote the number of ILC3s. Although we observed higher levels of *Il18*, we found equal or lower expression of all other putative epithelial factors that could control these cell types (*Il25, Il33, Tslp, Il15*) when comparing cKO epithelium with that from WT (Fig. S6E). Given that most of the factors known to influence ILC2s and ILC3s abundance were unchanged or reduced, we were prompted to assess other genes encoding cytokines and chemokines capable of interacting with ILC2s and ILC3s and controlled by LSD1, in an unbiased manner. In a previous study, we performed H3K4me1 ChIP-seq on small intestinal crypts from WT and cKO animals (note that these cKO animals did not have complete *Lsd1* deletion[15]). Here, we assessed this data set in search for significantly increased H3K4me1 levels in or near genes encoding for a cytokine or chemokine and combined this with receptor expression patterns in ILC2s and ILC3s from our scRNA-seq. We found increased levels of H3K4me1 near the gene that encodes for the chemokine CXCL16 (*Cxcl16*) (Fig. 6E), suggesting a possible effect of lack of demethylation by LSD1. In support, *Cxcl16* was expressed at significantly higher levels (~4-fold) in the intestinal epithelium of cKO mice compared to that of WT mice, particularly in villus epithelium and irrespective of ABX treatment (Fig. 6F and Fig. S6F). CXCL16 is a ligand for CXCR6, and, indeed, *Cxcr6* was expressed in our populations of interest (ILC2s and ILC3s) among other cell types (Fig. 6G).

Finally, in microbiota-depleting conditions (ABX-treated mice), we assessed what genes were differentially expressed in ILC2s and ILC3s by genotype using the scRNA-seq data set. Although this was done based only cells derived from a single mouse in each sample, we identified 140 genes that were commonly upregulated in both ILC2s and ILC3s derived from the lamina propria of cKO compared to WT ABX-treated mice (Fig. S6G), underscoring the impact of epithelial maturation in influencing the cellular signaling events of these ILCs. This overlap in upregulated genes in these two different cell types suggests that this could be due to a common activation mechanism, such as through the CXCL16-CXCR6 axis. Indeed, we found that many of the commonly upregulated genes are associated with processes canonically activated downstream of CXCL16-CXCR6 signaling, such as actin cytoskeletal reorganization and subsequent chemotaxis, PI3K-Akt, MAPK cascade, and RHO GTPase signaling[58] (Fig. 6H). To conclude, a detailed assessment of these 140 genes unearthed a network of Rho/Rac GTPases, actin cytoskeleton regulators and integrins commonly found in cell adhesion and migration processes (Fig. 6I). Although the sample size limits our ability to draw conclusions with regard to the gene network changes in ILCs, these results point to enhanced recruitment of ILC2s and ILC3s to the lamina propria of cKO mice, a process putatively driven by aberrant production of CXCL16 by cKO epithelium.

### ILC2s and ILC3s rapidly accumulate upon intestinal epithelial *Lsd1* deletion in adult mice

Taking advantage of our inducible cKO system (icKO, Fig. 2C), we tested if reverting matured epithelium into a neonatal-like state in adult

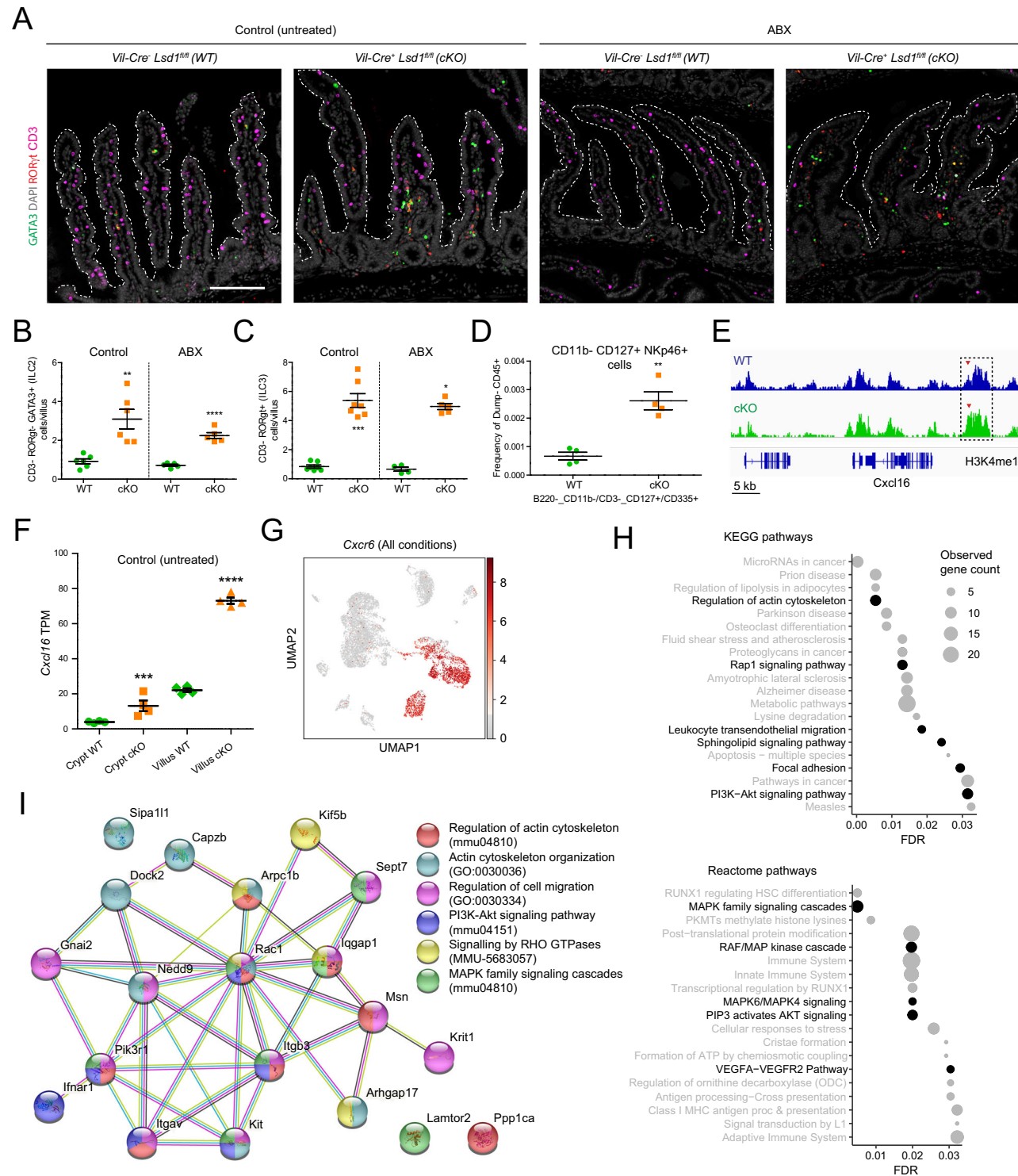

animals would affect ILC2s and ILC3s in the small intestine. Strikingly, as early as 2 days after 5 daily consecutive tamoxifen administrations by oral gavage (wk1 timepoint, see Fig. 2), we observed an increase in ILC2s and ILC3s in the lamina propria of icKO mice as compared to WT animals that received the same treatment (Fig. 7A, B and Fig. S7A). The increase in ILC2s and ILC3s in icKO animals was persistent throughout the 6-week experiment (Fig. 7A, B and Fig. S7A). Of note, these fast changes in cell numbers take place at a time when the fecal microbiota of icKO mice is practically unchanged as compared to that of a WT, and when there are no signs of diarrhea, pointing towards the proposed microbiota-independent hypothesis (see icKOwk1, Fig. 2H).

## Anti-CXCL16 treatment blocks ILC2 but not ILC3 accumulation upon intestinal epithelial *Lsd1* deletion in adult mice

To test if the hypothesized CXCL16-CXCR6 axis was the main contributing factor governing the increase of ILC2s and ILC3s in the lamina propria of cKO and icKO mice, we devised a signaling blockade experiment (Fig. S7B) using a monoclonal antibody targeting CXCL16 (αCXCL16). While ILC2 numbers were significantly affected by CXCL16 blockade (Fig.7C, D), ILC3s still increased after blocking CXCL16 (Fig. 7C, E) in icKO mice. Of note, we did not observe any changes in WT animals receiving the antibody treatment, suggesting that short-term ILC2/3 maintenance in adulthood is not mediated by

**Fig. 6 | LSD1-mediated intestinal epithelial maturation is required for the establishment and maintenance of ILC2s and ILC3s in a microbiota-independent manner. A** Type 2 and 3 Innate Lymphoid Cells (ILC2s and ILC3s) detection by immunofluorescence in mouse duodenum of WT and cKO mice (untreated and ABX-treated). ILC2 cells are defined as CD3⁻ RORγt⁻ GATA3⁺ while ILC3s are CD3⁻ RORγt⁺. Nuclei are counterstained with DAPI (gray). Villus structure is delimited by a discontinuous white line; Scale bar: 200 μm. See Fig. S6B for split channels. **B** Quantification of ILC2s in the lamina propria of the duodenum from untreated and ABX-treated mice (derived from Fig. 6A images). Data were presented as mean ± SEM; *n* = 6 mice/genotype (untreated) and 5 mice/genotype (ABX-treated) from two independent experiments (Two-tailed unpaired *t*-test). **C** Quantification of ILC3s in the duodenal lamina propria of untreated and ABX-treated mice (derived from Fig. 6A images). Data were presented as mean ± SEM; *n* = 7 mice/genotype (untreated) and 4WT/5cKO mice (ABX-treated) from two independent experiments (Two-tailed Mann–Whitney non-parametric test). **D** Flow cytometry data derived from small intestinal lamina propria CD45+ cells showing Ly6g⁻/B220⁻ CD11b⁻/CD3⁻ CD127⁺[IL-7Rα⁺]/CD335⁺[NKp46⁺] frequency. Data were presented as mean ± SEM; *n* = 4 mice/genotype from four independent experiments (Two-tailed unpaired *t*-test). Full gating hierarchy and strategy is shown in

(Fig. S4C–E). **E** Representative Integrative Genomics Viewer (IGV) tracks H3K4me1 levels near the *Cxcl16* locus (*n* = 2). Box indicates a significantly increased peak. **F** Bulk RNA-seq of crypt and villus fractions derived from WT and cKO untreated 2-month-old mice. Individual graphs show transcripts per million (TPM) counts. Data were presented as mean ± SEM; *n* = 4 mice/genotype, (Differential expression analyzed using DESeq2's negative binomial generalized linear model, with Benjamini–Hochberg adjusted *p* values). **G** Lamina propria CD45+-derived UMAP showing *Cxcr6* gene expression across all experimental conditions. Gray to red heatmap scale shows log(CPM+1). **H** Enrichment analysis of the 140 genes commonly upregulated in ILC2s and ILC3s derived from ABX-treated cKO mice and performed with the STRING online tool. The top 20 pathway hits with the lowest false discovery rate (FDR) are displayed. Canonical pathways activated by the CXCL16-CXCR6 axis are highlighted in black. **I** StringDB-derived interactome showing a 20 gene subset (of first and second-order Rac GTPase interactors) from the 140 gene list described in Fig. 6H and S6G. Participation of each gene in a particular biological process is shown as a color overlay (GO, MMU, and mmu prefix codes correspond to Gene Ontology: Biological Processes, Reactome pathways, and KEGG pathways databases, respectively). For interaction color coding see M&Ms. Source data and exact *p* values are provided as a Source Data file.

CXCL16, which is in line with work by others[59]. Our study thus indicates that LSD1-mediated repression of *Cxcl16* is necessary to limit ILC2 numbers. In contrast, ILC3 abundance might be controlled by other concomitant factors.

## Discussion

We show here that LSD1 is required for the postnatal maturation of epithelium that lines the small and large intestines (Figs. 1–3). Postnatal intestinal epithelial maturation comprises the appearance of Paneth cells and expansion and/or maturation of all other cell lineages, including goblet cells and enterocytes. This role for LSD1 seems tissue-specific, as it opposes its role in skin epithelium where inhibition of LSD1 led to the promotion of differentiation[60]. The microbiota can modulate intestinal epithelial cell biology, including Paneth cell numbers, for example, through the production of metabolites altering histone de-acetylase activity[39,61]. However, we find that LSD1 has an intrinsic and dominant role, as ABX treatment in vivo or culturing epithelium in vitro in organoids did not overtly affect the impaired intestinal epithelial maturation in the absence of LSD1 (Fig. 3). Using transcriptomic analysis, we find that adult *Lsd1*-deficient small intestinal epithelium is enriched for genes highly expressed at P7 and repressed by P21, the weaning stage (Figs. 1I, 3H). However, we do observe normal crypt formation, which is initiated in the first week of life, suggesting that up to P7, LSD1 does not play a major role in intestinal epithelial development. In contrast, the polycomb repressive complex 2 (PRC2) member EED (Embryonic ectoderm development) is necessary to repress genes normally expressed at the embryonic stage, thus prior to the formation of gut epithelium in development[14]. Indeed, deletion of EED in intestinal epithelium renders adult mice moribund within 2 weeks after the first tamoxifen administration, including the loss of proliferative crypt structures[14]. Thus, where EED is responsible for the mesoderm-to-epithelial transition, LSD1 is needed for the postnatal maturation transition. Together, epigenetic modifiers are strongly involved at different stages in intestinal epithelial development, and epigenetic enzymatic activity can be microbiota-dependent and independent.

The intestinal epithelium is an important modulator of the commensal microbial composition. Goblet cells produce mucins to form a protective layer, Paneth cells secrete antimicrobials such as lysozyme and defensins, and enterocytes express *Reg3b* and *Reg3g* in a microbiota-dependent manner[29,62]. Expectedly, the microbiome of adult cKO animals was different from its WT littermates (Fig. 3A, B). It is likely that some of the changes in the microbiome (e.g., Bacteriodaceae:Muribaculaceae ratio), as well as its severity, were influenced by the incidence of diarrhea in cKO mice. In support, more subtle

changes in the microbiome were found when deleting *Lsd1* later in life (icKO mice), which had much lower diarrhea incidence (Fig. 2F–I). Indeed, we propose that malabsorption causes osmotic diarrhea in cKO mice, rather than being a consequence of inflammation and potentially due to the dysbiosis of the microbiota. Accordingly, we found that diarrhea was also present in ABX-treated cKO mice (Fig. 3C), and thus, diarrhea occurred independently of the microbiota. In addition, we did not find evidence of increased inflammation in cKO mice. This includes uncompromised integrity of the intestinal barrier in cKO mice (Fig. S1H), no obvious immune cell recruitment in tissue sections, and the fact that primary and secondary lymphoid organs lacked signs of systemic inflammation (Figs. 4 and S4).

Besides its barrier function, the intestinal epithelium is intertwined with local immune cells to maintain gut homeostasis. Using scRNA-seq, we tested what immune cell types were microbiota-dependent and, by ABX treatment, what aspects of such immune cell composition were defined by LSD1-controlled epithelial maturation (Figs. 4–7). We found that IgA-producing plasma cell numbers were controlled by both epithelial maturation status and the microbiota (Fig. 5A–C). Goblet and M cells are involved in the transfer of antigen from the lumen[22–24], and this is likely to be affected in cKO mice. In addition, although the macrophage cellular state, as inferred by UMAP clustering, was mostly unaffected, we determined that the establishment or maintenance of a CD163+ macrophage population was dependent on intestinal epithelial maturation. Potentially, via the crypt-enriched expression of chemokines *Ccl6* and *Ccl9* by goblet and a small subset of Paneth cells (Fig. 5H and Fig. S5D), which are ligands for macrophage-expressed CCR1 (Fig. 5G). Furthermore, both the cellular state as well as the number of ILC2s and ILC3s in the lamina propria are controlled by epithelial-expressed *Lsd1*. We propose that LSD1 normally represses *Cxcl16* as we found heightened levels of H3K4me1 near its locus (Fig. 6E) and strongly increased *Cxcl16* expression in cKO epithelium compared to WT epithelium (Fig. 6F and Fig. S6F). ILC2s and ILC3s express *Cxcr6*, the CXCL16 receptor (Fig. 6G), and thus we initially hypothesized that heightened *Cxcl16* levels could cause both ILC2 and ILC3 recruitment or expansion. Indeed, we find that a gene set upregulated in both ABX cKO-derived ILC2s and ILC3s is associated with pathways activated by a CXCL16-CXCR6 axis (Fig. 6H, I). In support, mice lacking *Cxcr6* have reduced numbers of ILC3s in homeostasis[63] and the CXCL16-CXCR6 axis has been shown to govern migration in ILC2s[64–66]. In our study, the changes in ILC numbers occurred quite rapidly as deletion of *Lsd1* in adult epithelium led to an increase of ILC2s and ILC3s within a week (Fig. 7A, B and Fig. S7A). Of note, CXCL16 expression is normally associated with dendritic cells[63,67], however, as the epithelium is by far the most prevalent cell

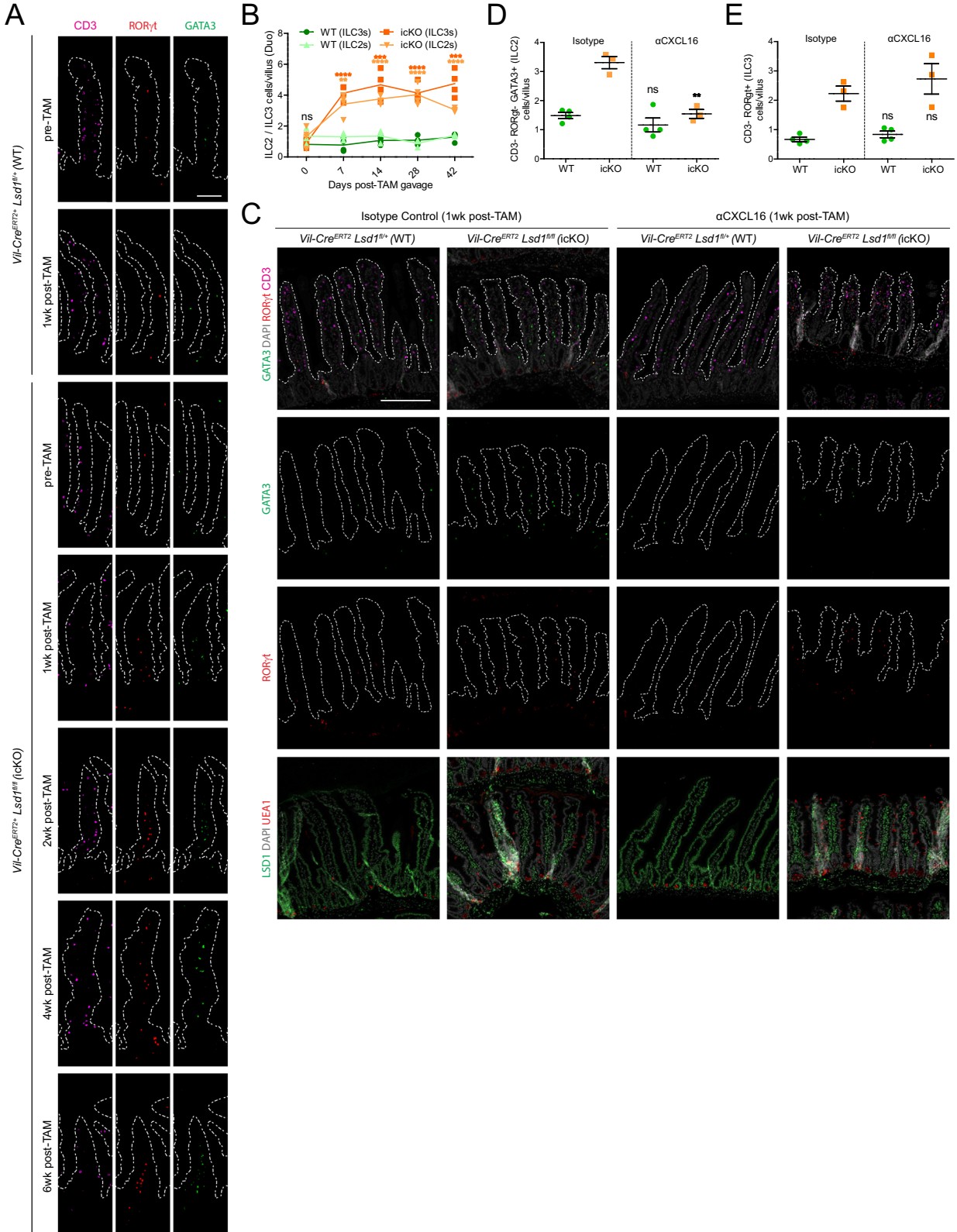

type in the gut, increased expression could lead to accumulating protein levels that are epithelial derived.

Finally, we use αCXCL16 in vivo to provide evidence that the CXCL16-CXCR6 axis is a key driver behind the ILC2 accumulation upon *Lsd1* deletion in the intestinal epithelium. In contrast, ILC3 accumulation was unaffected in αCXCL16-treated mice (Fig. 7). While most epithelial-derived factors normally associated with controlling ILC2

and ILC3 cell numbers were not strikingly increased and therefore unlikely contributors (Fig. S6E), *Il18*, a cytokine described to drive ILC3 proliferation[56], was significantly upregulated in cKO intestinal epithelium (Fig. S6E). This would potentially allow ILC3s to circumvent the effects of CXCL16 neutralization via the IL18-IL18R pathway.

The results of this study confirm that LSD1 is not only necessary for the correct maturation of the intestinal epithelium but also for the

**Fig. 7 | Rapid induction of CXCL16-dependent ILC2s and CXCL16-independent ILC3s upon epithelial-intrinsic *Lsd1* deletion in adult mice. A** ILC2s and ILC3s detected by immunofluorescence in mouse duodenum of WT and icKO mice; before tamoxifen treatment (pre-TAM) and after 1 to 6 weeks of initial tamoxifen dosage (post-TAM 1-6wk). ILC2 cells are defined as CD3⁻ RORγt⁻ GATA3⁺ while ILC3s are CD3⁻ RORγt⁺. Nuclei are counterstained with DAPI (gray). Villus structure is delimited by a discontinuous white line; Scale bar: 100 μm. See Fig. S7A for a zoomed-out representative section. **B** Quantification of ILC2s and ILC3s in the lamina propria of duodenums from pre- and post-TAM treated mice (derived from Fig. 7A images). Data were presented as individual data points in a time series; *n* = 4 mice/genotype/timepoint (Individual timepoints/cell type assessed by two-tailed unpaired *t*-test). **C** ILC2s and ILC3s detected by immunofluorescence in mouse duodenum of 1wk post-TAM WT (subject to Isotype control or αCXCL16 treatment,

as shown in Fig. S7B). ILC2 cells are defined as CD3⁻ RORγt⁻ GATA3⁺ while ILC3s are CD3⁻ RORγt⁺. Nuclei are counterstained with DAPI (gray). Villus structure is delimited by a discontinuous white line; Scale bar: 200 μm. **D** Quantification of ILC2s in the lamina propria of the duodenum from Isotype control and αCXCL16-treated mice (derived from Fig. 7C images). Data were presented as mean ± SEM; *n* = 4 mice/genotype (Isotype control) and 3 mice/genotype (αCXCL16-treated), (Two-tailed unpaired *t*-test, comparisons across treatments). **E** Quantification of ILC3s in the lamina propria of the duodenum from Isotype control and αCXCL16-treated mice (derived from Fig. 7C images). Data were presented as mean ± SEM; *n* = 4 mice/genotype (Isotype control) and 3 mice/genotype (αCXCL16-treated), (Two-tailed unpaired *t*-test, comparisons across treatments). Source data and exact *p* values are provided as a Source Data file.

maintenance of this mature status. In addition, we show a new gut microbiota-independent axis of communication between the intestinal epithelium and the immune cell compartment, a process directly influenced by the maturation status of said epithelium. Finally, although this study provides the backbone for direct communication between the intestinal epithelium and several immune cell types, further intervention-type work on ligand–receptor interactions blockade is needed to characterize these in more depth.

## Limitations of the study
In our work, we continuously used litter and cage mate controls, which is important, especially when studying intestinal biology and gut microbiota[68]. However, as mice are coprophagic, it also means that mice continuously influence the microbiota of cage mates, which is especially relevant for the induced deletion of *Lsd1* experiment and the analysis of the microbiome (Fig. 2). For example, we would imagine a stronger or more rapid effect of *Lsd1* deletion on the microbiome in the absence of WT littermates. In addition, much of our work relies on transcriptomics, staining of tissue sections, and flow cytometry. This approach may overlook potential functional alterations, such as the ability of epithelium to absorb nutrients or the effect epithelial-derived factors have on the function of immune cells, such as phagocytosis or their cytokine levels. In addition, our approach was limited in the extent to which we could define immune cell populations. For example, there are several subsets of ILC2s and ILC3s, which we did not quantify or qualify in detail.

## Methods
### Animal experiments
**Ethics statement.** Mice were maintained at the Comparative Medicine Core Facility (CoMed) at NTNU (Norges Teknisk Naturvitenskaplige Universitet) in accordance with the Norwegian Guidelines for Animal Research. Antibiotic cocktail and tamoxifen-induced deletion experiments were assessed and approved by the Norwegian Food Safety Authority (FOTS ID 21275).

**Mouse strains.** *Villin*-Cre (Jackson Laboratories, Strain #: 021504, RRID:IMSR_JAX:021504), *Villin*-CreERT2[27] (kind gift from Dr. Robine), and *Lsd1*f/f [69] (kind gift from Stuart Orkin) mice were housed in CoMed. *Villin*-Cre *Lsd1*f/f (cKO) mice were housed under specific-pathogen-free (SPF) conditions, and *Villin*-CreERT2 *Lsd1*f/f (icKO) mice were maintained in the minimal disease unit at CoMed. Mice were housed with controlled temperature between 21 and 22 degrees Celsius and relative humidity between 45 and 60%. The animals are housed in a 12-h dark/12-h light cycle, with 1 h of dusk/dawn. All mice used were between 8–14 weeks of age. By conventionally bred or conventionally raised animals, we refer to mice (of all genotypes) housed in these conditions and not treated with antibiotics.

**Mice euthanasia method.** Euthanasia of rodents was done via carbon dioxide ($CO_2$) inhalation. In order to cause minimal distress and obtain

rapid unconsciousness mice were euthanized in their home cage A fill rate of 30–70% of the chamber volume per minute with 100% $CO_2$, added to the existing air in the chamber, was used to achieve a balanced gas mixture.

**Antibiotic cocktail treatment (ABX).** Parental mice were treated with an antibiotic cocktail (ABX) in drinking water 2 weeks prior to mating and subsequent offspring kept being administered ABX for the duration of the experiment. ABX was freshly prepared weekly, kept at 4 °C, and changed every 2–3 days. Antibiotic cocktail (ABX): Ampicillin (A0166-5g, Sigma-Aldrich) (1 g/liter), Amoxicillin-Clavulanate (SMB00607-1G, Sigma-Aldrich) (0.25 g/liter), neomycin (Colivet vet., VetPharma AS) (1 g/liter), gentamicin (G1397-100 ml, Sigma-Aldrich) (1 g/liter), and vancomycin (V2002, Sigma-Aldrich) (0.5 g/liter) in drinking water. Sucralose (sweetener) was added at 0.1 g/liter to make the cocktail palatable.

**Tamoxifen administration.** Tamoxifen (Sigma-Aldrich, T5648-1G) was dissolved in 50 ml of Corn Oil (Sigma-Aldrich, C8267-500 ML) at a 20 mg/ml concentration and stored at 4 °C. To induce *Lsd1* recombination, 8–12-week-old mice were administered daily via oral gavage with 0.1 ml of Tamoxifen for a duration of 5 days.

**CXCR6-CXCL16 signaling blockade experiment (αCXCL16)**
To induce *Lsd1* recombination, 8–12-week-old mice were administered daily via oral gavage with 0.1 ml of Tamoxifen for a duration of 5 days. Monoclonal rat IgG2A CXCL16 neutralizing antibody (R&D Systems, Clone #142417, MAB503) or monoclonal Rat IgG2A isotype control (R&D Systems, Clone #54447, MAB006) was administered via intra-peritoneal injection at a dose of 100 μg/mouse (in 200 μl of sterile PBS) for 4 alternating days, starting the day before the first tamoxifen oral gavage. Antibodies were freshly reconstituted in sterile PBS on the day of the injection. See Fig. S7B for the treatment scheme.

### Intestinal epithelium isolation
**Crypt and villi isolation.** Briefly, we isolated 10 cm from the duodenal section of mice intestine. The duodenum was washed with ice-cold PBS, opened longitudinally, and cleaned of mucus with a coverslip. Next, villi were scraped off using a glass coverslip and collected with ice-cold PBS. Collected villi were filtered through a 70-μm cell strainer and cleaned with 20 ml of ice-cold PBS. By inverting the filter, villi were recovered using PBS and centrifuged at 300×*g* for 5 min. Villi fractions were snap-frozen in liquid nitrogen for subsequent RNA extraction. Afterward, we followed the standard protocol for crypts isolation as described[70]. The remaining duodenum tissue was cut into 2–4 mm small pieces and transferred to a 50 mL tube. These fragments were washed with ice-cold PBS up and down with a 10 mL pipette coated with FBS. This step was repeated ten times until the supernatant turned fully clear. Next, these fragments were incubated with 30 mL ice-cold crypt isolation buffer (2 mM EDTA in PBS) with gentle rotation in the cold room. After 30 min, fragments were settled down and the

supernatant was discarded. After this, fragments were vigorously shaken with 20 ml of PBS to release the villus and crypts. This step was repeated until most of the crypts were released and passed through the 70-μm cell strainer and collected into FBS coated sterile 50 ml falcon tube. The filtered fractions contained crypts and were spun down at 300 × g for 5 min, and the pellet was resuspended in 10 mL of ice-cold basal culture media. This fraction was washed with ice-cold basal culture media to remove single cells at 200 × g for 2 min. Next, pellets were either snap-frozen for RNA extraction or resuspended and counted under a light microscope for seeding as an organoid culture.

**Organoid cultures.** Organoids were generated by seeding ca. 250–500 small intestinal crypts in a 40-μl droplet of cold Matrigel (#734-1101, Corning) into the middle of a prewarmed 24-well plate. Matrigel was solidified by incubation at 37 °C for 5–15 min and 500 μl of basal culture medium (ENR) was added. Basal culture medium ("ENR") consisted of Advanced DMEM F12 (Gibco) supplemented with 1x Penicillin-Streptomycin (Sigma-Aldrich), 10 mM HEPES, 2 mM Glutamax, 1x B-27 supplement, 1x N 2 supplement, (all Gibco) 500 mM *N*-acetylcysteine (A7250, Sigma-Aldrich), 50 ng/ml recombinant EGF (PMG8041, Thermo Fisher Scientific), 10% conditioned medium from a cell line producing Noggin (kind gift from Hans Clevers), and 20% conditioned medium from a cell line producing R-Spondin-1 (kind gift from Calvin Kuo). ENR culture medium was replaced every 2–3 days. Organoids were passaged at a 1:3–1:4 ratio by disruption with rigorous pipetting almost to single cells. Organoid fragments were centrifuged at 300 × g, resuspended in 40-μl cold Matrigel per well, and plated on prewarmed 24-well plates.

**Generation of tamoxifen-induced *Lsd1*$^{-/-}$ organoids.** Crypts were extracted from *Villin*-CreERT2 *Lsd1*$^{f/f}$ (icKO) and *Villin*-CreERT2 *Lsd1*$^{f/+}$ (WT) mice and seeded as described under Organoid cultures above. At 48-h post-seeding, organoids were treated with 1 μg/ml tamoxifen (T5648, Sigma-Aldrich) to induce Cre recombinase-mediated deletion of Lsd1. Tamoxifen was reconstituted in ethanol to produce a 10 mg/ml stock solution and an equivalent ethanol dilution (1:10,000) was prepared in an ENR medium for control-treated organoids. At 24-h post-treatment (and 72-h post-seeding), culture medium was replaced with fresh ENR medium and organoids were cultured for an additional 3 days before passaging. Organoids were imaged and RNA was extracted 4 days after this first passage.

**Fluorescence confocal microscopy staining, imaging, and quantification**

**Immunofluorescence staining of paraffin-embedded tissue and organoids.** After sacrificing the mice, 10 cm of representative sections of each intestinal tract portion (Duodenum, Jejunum, Ileum, and colon) were dissected, washed with ice-cold PBS, and rolled into "swiss rolls" as previously described[71]. Intestinal tissue was fixed with 4% formaldehyde for 24 h at RT (room temperature). After fixation, tissues were paraffin-embedded, cut into 4-μm sections, and placed in slides. Tissue slides were then deparaffinized and rehydrated, followed by antigen retrieval (Citrate buffer pH-6.0 for all stainings). Sections were blocked in blocking/permeabilization buffer (0.2% Triton X-100, 1% BSA, and 2% normal goat serum (NGS) in 1X PBS) for 1 h at RT in a humidified chamber. Tissues were then incubated overnight at 4 °C in a staining buffer (0.1% Triton X-100, 0.5% BSA, 0.05% Tween-20, and 1% NGS in PBS) with diluted primary antibodies: (LSD1, 1:200, Cell Signaling Technology, Cat. No. 2184S), (MUC2, 1:200, Santa Cruz Biotechnology, sc-15334), (LYZ, 1:200, Agilent DAKO, A0099), (β-CAT, 1:200, BD Biosciences, 610154), (CD3, 1:200, Novusbio, NB600-144155), (RORγt, 1:200, Thermo Fisher, 12-6988-82), (GATA3, 1:200, Abcam, ab199428), (CD163, 1:200, Abcam, ab182422), (CD68, 1:200, BioRad, MCA1957T), (DCAMKL1/DCLK1, 1:200, Abcam, ab31704), (E-Cadherin (24E10), 1:200, Cell Signaling, 3195) and (IgA, 1:200,

Novusbio, NB7503). The next day, slides were washed with 0.2% Tween-20 in PBS. After three washing steps, tissues were incubated with secondary antibodies coupled to fluorochromes, UEA1 (1:500, Vector Laboratories, RL-1062), and counterstained with Hoechst 33342 (1:1000, MERCK, H6024) for 1 h at RT. After incubation, slides were washed three times with 0.2% Tween-20 in PBS and then mounted in Fluoromount G (Invitrogen, 00-4958-02). For signal amplification of CD163 and multiplexing with primary antibodies from the same species (Rabbit, CD3, and GATA3), the Tyramide SuperBoost™ (Thermo Fisher, B40922) kit was used. For organoid staining we followed an already published protocol[17]. In short, organoids were seeded in pre-warmed eight-well chamber IBIDI slides. After incubation and growth, organoids were fixed in 4% paraformaldehyde with 2% sucrose for 30 min. Wells with fixed organoids were washed twice with sterile PBS for 5 min each. Next, organoids were permeabilized with 0.2% Triton X-100 prepared in PBS and incubated for 30 min at room temperature. To block free aldehydes groups and prevent high background signal, organoids were incubated with 100 mM glycine for 1 h at room temperature. The organoids were then incubated in the blocking buffer (1% BSA, 2% NGS diluted in 0.2% Triton X-100 in PBS) in a humidified chamber for 1 h at room temperature. Organoids were incubated in the diluted primary antibodies (LSD1, 1:200, Cell Signaling Technology, Cat. No. 2184S) and (β-CAT, 1:200, BD Biosciences, 610154) in the same blocking buffer overnight at 4 °C. Next day, the primary antibody solution was removed by pipetting, and wells were washed three times for 10 min each with 0.2% Tween-20 in PBS with slight agitation. Organoids were incubated with secondary antibodies, UEA1 (1:500, Vector Laboratories, RL-1062), and counterstained with Hoechst 33342 (1:1000, MERCK, H6024) overnight at 4 °C in the dark. Next, the secondary antibody solution was removed by pipetting, and wells were washed three times with 0.2% Tween-20 in PBS for 10 min in agitation. After staining, wells were mounted in 250 μL of Fluoromount G and were visualized with a confocal microscope. The following secondary antibodies from Invitrogen (Thermo Fisher Scientific) were used at a 1:500 dilution for immunostaining: (Goat anti-Rabbit 488, A-11034), (Goat anti-Mouse 555, A-21422), (Goat anti-Rabbit 555, A-21428), (Goat anti-Rat 555, A-21434), (Goat anti-Mouse 647, A-21237), (Goat anti-Rabbit 647, A-21245), and (Goat anti-Rat 647, A-21247).

**RNAScope (in situ RNA analysis).** To perform the 4-plex in situ hybridization on pretreated formalin-fixed paraffin-embedded tissues both RNAScope Multiplex Fluorescent Kit v2 (ACD Biotechne, Cat. No. 323100) and 4-Plex Ancillary Kit (ACD Biotechne, Cat. No. 323120) kits were used in combination with Opal reagents from Akoya Biosciences (Opal 520−Cat. No. FP1487001KT, Opal 570−Cat. No. FP1488001KT, Opal 620−Cat. No. FP1495001KT, Opal 690−Cat. No. FP1497001KT) for detection of fluorescent signals. The following probes were used for target detection: (Mm-Ada-O1, Cat. No. 832851), (Mm-Nt5e, Cat. No. 437951), (Mm-Lyz-C2, Cat. No. 415131-C2), (Mm-Muc2-C3, Cat. No. 315451-C3), and (Mm-Reg1-C4, Cat. No. 511571-C4). Tissue processing and staining workflow was carried out according to the RNAScope 4-plex Ancillary Kit for Multiplex Fluorescent Reagent Kit v2 technical note.

**Confocal microscopy imaging.** Images were obtained on a Zeiss LSM880 Airyscan using two air objectives: Plan-Apochromat 10x/0.45NA/2mm WD (420640-9900) and a Plan-Apochromat 20x/0.8NA/0.55mm WD (420650-9902).

**Image acquisition, processing software, and quantification.** Images were acquired using Zen 2.3 Black and gray levels/maximum intensity projections were adjusted in Zen 2.3 Blue prior to .tiff export. Cell counts were manually quantified in an unbiased and blind manner. Researchers were blind to sample genotype and treatment during tissue collection, processing, staining, imaging, and quantification using sample numbers as identifiers. To quantify cells, we randomly

selected complete and intact villi across the entire Swiss roll. Based on availability, 25–30 morphologically complete villi were counted and averaged per biological replicate (one tissue section per biological replicate). After quantification, each sample number was cross-referenced to genotype and treatment. Depicted scale bars apply to all images within the same panel.

### Lamina propria, spleen, and mesenteric lymph node leukocyte isolation

After sacrificing 2.5 month old mice, lamina propria leukocyte isolation (from the entire small intestine length) was done according to the referenced method[72] with a few modifications, namely: HBSS was substituted with DPBS (w/o $CaCl_2$ and $MgCl_2$) and the tissue digestion was performed on 15 ml of prewarmed (37 °C) RPMI 1640 containing: 10% FCS + Pen/Strep (10 mM) + Glutamine (2 mM) + Collagenase D 7.5 mg + Dispase 45 mg, and DNAse 7.5 mg + 100 ul Collagenase VIII (62.5 CDU/ml). For spleen isolations, the spleen was minced using scissors and then ground using the flat end of a syringe in 5 ml of RPMI on a 100-ml culture dish. The resulting cell suspension was passed through a 40-um cell strainer into a 50-ml tube. Cells were spun down, the supernatant removed, the cell pellet resuspended in Red Blood Cell Lysis Buffer (eBioscience, Cat. No. 00-4333-57), and incubated for 2 min at RT. Afterward, the lysis reaction was diluted with 45 ml of RPMI, cells spun down and resuspended in PBS + 10% FBS for surface staining. Mesenteric lymph nodes were harvested, with care to remove any remaining fat, and stored in cold PBS with 10% FCS. Harvested lymph nodes were passed through a 70-µm cell strainer to obtain a single-cell suspension. Cell numbers and viability were assessed using Trypan blue in an EVE automated cell counter (VWR).

### Flow cytometry

Single cells were stained with Zombie Aqua (Biolegend, 1:1000 in PBS) for 15 min at room temperature (RT) for live-dead exclusion. Samples were incubated with two different panels (St1 and St2), consisting of antibody conjugates against [St1: CD335-BV421, CD3-BV605, CD127-BV711, CD8-BV785, CD25-AF488, TCRgd-PerCp-Cy5.5, TCRb-PE, CD4-APC, CD45-APC-Fire, Dump (CD326, CD19, CD11b, Ly6g, Ter119)-PE-Cy7] and [St2: CD335-BV421, CD3-BV605, CD127-BV711, CD25-BV785, MHCII-AF488, CD11c- PerCp-Cy5.5, B220-PE, CD11b-AF647, CD45-APC-Fire, Dump(CD326, Ly6g, Ter119)-PE-Cy7] [all Biolegend, 1:200 in PBS + 2% fetal calf serum (FCS)] for 20 min at 4 °C. All the samples were analyzed using a BD LSR II flow cytometer (BD Biosciences) equipped with 405, 488, 561, and 647 nm laser lines. Single fluorochrome stainings of cells and compensation particles (BD CompBead, Becton Dickinson) were included in each experiment. For analysis, FlowJo software v10.6.2 and GNU R/Bioconductor v3.6.3/v3.10 packages flowCore, CytoML/flowWorkspace, ggcyto, and flowViz were used.

### Nucleic acid extractions, sequencing, and analysis

**RNA extraction and qPCR analysis.** RNA was isolated from organoids using the RNeasy Plus Micro Kit (Qiagen, Cat. No. 74034) and from intestinal epithelium (crypts and villi) using the RNeasy Plus Mini Kit (Qiagen, Cat. No. 74134) in combination with QIAshredder columns (Qiagen, Cat. No. 79656). All samples were snap-frozen in liquid nitrogen on isolation day and purified simultaneously later. RNA was quantified by spectrophotometry (ND1000 Spectrophotometer, NanoDrop). cDNA was synthesized from 1 µg of total RNA using the High-Capacity RNA-to-cDNA™ Kit (Applied Biosystems, Cat. No. 4387406). qPCR was performed using Fast SYBR Green Master Mix (Applied Biosystems, Cat. No. 4385616) in a StepOnePlus™ Real-Time PCR System (Applied Biosystems). All samples were analyzed in triplicate, and RNA levels were obtained with StepOnePlus™ Software v2.3 (Applied Biosystems). For fold change expression normalization, mGAPDH and mHPRT housekeeping genes were used. See Table S1 for a list of primers used in this study.

### gDNA extraction from mouse stool and qPCR assessment of relative bacterial load

Genomic DNA extraction from frozen stool (snap-frozen in liquid N2) samples was done using the QIAamp Fast DNA Stool Mini Kit (Qiagen, Cat. No. 51604) according to the manufacturer's protocol. Lysis temperature was adjusted to 95 °C. Enumeration of total bacterial load by qPCR was achieved by amplification of the 16S rRNA genes, as previously described[73], (primer pair 16S-341_F and 16S-805_R; CCTACGGGNGGCWGCAG and GACTACHVGGGTATCTAATCC, respectively) using a StepOne Real-Time PCR Detection System (Thermo Fisher) in 25 µl reactions containing 12.5 µl iQ SYBR Green Supermix (BioRad), 2 µl template DNA (1:1000 diluted), 300 nM of both primers 16S-341_F and 16S-805_R. The PCR amplification program consisted of an initial denaturation set at 95 °C for 3 min. followed by 35 three-step cycles at 95 °C for 15 s and at 55 °C for 20 s and 72 °C for 30 s. In each run, negative template controls (DNA replaced by nuclease-free water in qPCR) were included. Melting curves were checked for each sample to confirm the amplification of the correct product. The relative bacterial load of each sample was normalized to the initial stool sample weight.

### Bulk RNA-seq sequencing

Library preparation and sequencing were performed by Novogene UK using the NEB Next Ultra RNA Library Prep Kit (New England BioLabs, Cat. No. E7530L). Samples were sequenced using 150-bp paired-end reads using NovaSeq 6000 (Illumina).

### Bulk RNA-seq analysis

Reads were aligned to the *Mus musculus* genome build mm10 using the STAR aligner[74]. The count of reads that aligned to each exon region of a gene in GENCODE annotation M18 of the mouse genome[75] was counted using featureCounts[76]. Genes with a total count of less than ten across all samples were filtered out. Differential expression analysis was done with the R package DESeq2, and volcano plots were plotted with the R package EnhancedVolcano[77]. PCA analysis was performed with the scikit-learn package with the function sklearn.decomposition.PCA[78]. GSEA analysis was run with the log2(fold change) calculated by DESeq2 as weights, 10,000 permutations, and otherwise default settings using the R package clusterProfiler[79]. GO term analysis was run using the function enrichGO in clusterProfiler. Used gene sets in GSEA analysis are described in Supplementary Data 1. The R packages pheatmap and eulerr were used to make heatmaps and Venn diagrams, respectively.

### scRNA-seq sequencing and analysis

scRNA-seq was performed using 10X genomics Chromium Next GEM Single Cell 3′ GEM, Library & Gel Bead Kit v3.1 and sequenced using two Illumina NS500HO flowcells with a 75 cycle kit. Reads were aligned to the genome and reads per gene were counted using the cellranger count function (v3.1.0) with the cellranger reference mm10 3.0.0[80]. Samples were then combined with cellranger aggr. Scanpy (v1.5.1)[81] was used to quality filter the raw data with the following filters: min_genes_per_cell = 400, min_cells_per_gene = 3, max_counts_per_cell = 40,000, max_percent_mt_per_cell = 20. Raw reads were converted to counts per million and log-transformed. Cell neighbors were computed with 10 neighbors and 40 PCA dimensions. Leiden algorithm was used with a resolution of 1 to split clusters and UMAP to display them. CIPR with the ImmGen data set from mouse was used to initially identify the cell clusters[82], and these were confirmed with cell type markers. Scanpy was used to make plots of cell clusters.

### StringDB gene enrichment

Version 11.5 of STRING Database was used for gene enrichment and network graph display. Interaction sources to build up the network include: Text mining, experiments, databases, co-expression, neighborhood, gene fusion, and co-occurrence. Only experimentally determined (magenta) and known interactions (cyan) from curated

databases are depicted along interactions derived from text mining (green) and co-expression (black).

## Microbiome sequencing

16S rDNA Amplicon Metagenomic Sequencing of bacterial gDNA derived from mouse stool was outsourced to Novogene UK. The Amplicon library of the V4 region was obtained using the 515F (5′-GT GCCAGCMGCCGCGGTAA-3′) and 806R (5′-GGACTACHVGGGTWTC TAAT-3′) primer pairs connected with barcodes. The PCR products with proper size were selected by 2% agarose gel electrophoresis. The same amount of PCR products from each sample was pooled, end-repaired, A-tailed, and further ligated with Illumina adapters. Paired-end sequencing (PE250) was performed on the obtained libraries based on the Illumina NovaSeq sequencing platform.

## Microbial 16S amplicon analysis

Amplicon sequence analysis was outsourced to Novogene UK. Paired-end reads was assigned to samples based on their unique barcodes and truncated by cutting off the barcode and primer sequences. After the barcode sequence and primer sequence were truncated, FLASH (v1.2.11, http://ccb.jhu.edu/software/FLASH/)[83] was used to merge the reads to get raw tags. Then, fastp software was used to do quality control of raw tags, and high-quality clean tags were obtained. Finally, Vsearch software was used to blast clean tags to the database to detect the chimera and remove them, to obtain the final effective tags. For the effective tags, the DADA2 or deblur module in QIIME2 software was used to do denoise (DADA2 were used by default), and the sequences with less than 5 abundance were filtered out to obtain the final ASVs (Amplicon Sequence Variables) and feature table. Then, the Classify-sklearn moduler in QIIME2 software was used to compare ASVs with the database and to obtain the species annotation of each ASV. QIIME2 software was used to calculate alpha diversity indices, including observed_ otus, shannon, simpson, chao1, goods_ coverage, dominance, and pielou_e indices. The rarefaction curve and species accumulation boxplot were drawn. If there was grouping, the differences between groups of alpha diversity would be analyzed by default. For Beta Diversity analysis, the UniFrac distance was calculated by QIIME2 software, and the dimensionality reduction maps of PCA, PCoA and NMDS were drawn by R software. Among them, Ade4 and ggplot2 packages in R software were used to display PCA and PCoA. Then, Adonis and Anosim functions in QIIME2 software were used to analyze the significance of community structure differences among groups. Finally, LEfSe or R software was used to perform the species analysis of significant differences between groups. LEfSe analysis was performed by LEfSe software, and the LDA score threshold was set to 4 by default. In MetaStat analysis, R software was used to test the differences between the two groups at the level of phylum, class, order, family, genus and species, and $P$ value was obtained. The species with $P$ value less than 0.05 were selected as the significant differences between the two groups; in $T$-test, R software was also used to analyze the significant differences of species at each taxonomic level.

## Statistical analysis

Unless stated otherwise in the methods section, statistical analysis, and representation was carried out using Graphpad Prism V8.0.2. Prior to determining statistical significance, we assessed data distribution using the Kolmogorov–Smirnov (KS) and Shapiro–Wilk (SW) tests. In instances where the sample size was too small to confidently apply the KS test, the SW test was exclusively used. Normality was affirmed only when data passed both tests (where applicable), allowing for the subsequent application of parametric tests to analyze statistical significance. Conversely, for data that did not meet these criteria, non-parametric tests were utilized (One- or two-tailed Mann–Whitney). We excluded RNA-Seq data expression and microbial population analyses, as these datasets incorporate inherent checks for data distribution

within their analysis pipelines. Statistical significance was evaluated by the appropriate methods stated in the figure legends or in the methodology section. Means and individual data points (biological replicates) were represented as ±SEM unless stated otherwise. Differences were considered statistically significant at *$P < 0.05$, **$P < 0.01$, ***$P < 0.001$, and ****$P < 0.0001$. Exact $p$ values and tests applied can be found in both Source Data files. We have removed statistical significance from comparisons between the villus compartments for crypt-restricted markers like *Mki67*, *Lgr5*, *Olfm4*, *Lyz*, and *Defa22*, and vice versa for *Slc15a1*, *Cd36*, *Apob*, and *Npc1l1*, marking them as "n/a" (does not apply) to avoid potential data misinterpretation.

## Reporting summary

Further information on research design is available in the Nature Portfolio Reporting Summary linked to this article.

## Data availability

We provide a Source Data file for all the presented graphs (Source_Data_1.xlsx). For gene expression graphs derived from bulk RNA-Seq experiments, we supply an additional Source_Data_2_RNASeq_DiffExpression.xlsx. Both are compressed in a single Source Data .zip file. All sequencing data was uploaded via Annotare to the ArrayExpress repository at BioStudies EMBL-EBI (www.ebi.ac.uk/arrayexpress) under the following accession numbers. For bulk RNA-seq-derived data: E-MTAB-10498. (Crypt and villi small intestinal epithelium from untreated WT and cKO mice), E-MTAB-10499. (Crypt and villi small intestinal epithelium from ABX-treated WT and cKO mice) and E-MTAB-10500. (Passage number 4, day 4 organoids derived from the duodenum of untreated WT and cKO mice). For scRNA-Seq-derived data: E-MTAB-10492. For 16S rDNA Amplicon Metagenomic Sequencing of bacterial gDNA derived from mouse stool: E-MTAB-12489. (Adult vs Neonate) and E-MTAB-12626. (Timepoints before and after *Lsd1* tamoxifen-induced recombination in the intestinal epithelium). Raw imaging data and .fcs flow cytometry files can be shared on request. Source data are provided with this paper.

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

## Acknowledgements

We thank Drs. Stuart Orkin and Sylvie Robine for kindly sharing mouse strains. We thank Unni Nonstad and Liv Ryan for their assistance with cell sorting. We thank the imaging (CMIC) and animal care (CoMed) core facilities that assisted in this work (Norwegian University of Science and Technology, NTNU). The single-cell RNA-seq was done by the Genomics Core Facility at NTNU. Imaging was performed at the Cellular and Molecular Imaging Core Facility (CMIC) at NTNU. Genomics Core and CMIC receive funding from the Faculty of Medicine and Health Sciences and Central Norway Regional Health Authority. Funding for this work was provided by the Research Council of Norway (Centre of Excellence grant 223255/F50, and "Young Research Talent" 274760 to MJO and 326209 to M.M.-A.) and the Norwegian Cancer Society (182767 to M.J.O. and 245170 to M.M.-A.). T.N.S. was funded by the BBSRC (BB/S01103X/2).

## Author contributions

Conceptualization of the study: A.D.-S and M.J.O.; Performed experiments: A.D.-S., P.M.V., M.M.-A., J.O., R.Y., A.B.S., A.M., and N.P.; Analysis and interpretation of data: A.D.-S., M.M.-A., H.T.L., R.Y., A.B.S., T.N.S., and J.O.; Bioinformatic analysis: H.T.L.; Made figures: A.D-S., M.M.-A., H.T.L., and P.M.V.; Supervision: M.J.O.; Wrote first draft: A.D.-S., M.M.-A., and M.J.O. Contributed to final draft: A.D.-S., H.T.L., P.M.V., N.P., J.O., R.Y., A.B.S., A.M., T.N.S., M.M.-A., and M.J.O.

## Funding

## Competing interests

The authors declare no competing interests.
