## [Peer Review File · Nature Communications]

LSD1 drives intestinal epithelial maturation and controls small intestinal immune cell composition independent of microbiota in a murine modelREVIEWER COMMENTS

Reviewer #1 (Remarks to the Author):

In this study, Díez-Sánchez et al explore the intestinal epithelial-intrinsic role of the epigenetic modifier LSD1 in regulating postnatal epithelial maturation in the context of the developing luminal microbiota and sub-epithelial immune cell populations. They initially validate that conditional LSD1 KO (cKO) mice have altered epithelial maturation along different regions of the GI tract, and that cKO mice display reduced post-weaning weight gain and diarrhoea. Furthermore, they identify striking divergence in post-weaning microbiota development; however, via application of antibiotic (ABX)-mediated microbiota suppression they illustrate that epithelial maturation deficiency in cKO mice is independent of the microbiota. They go on to explore the relative roles of the microbiota and LSD1 in regulating mucosal immunity by profiling lamina propria immune cell populations by single cell RNA sequencing (scRNA-seq), identifying several alterations with potential biological relevance. They finally focus on LSD1-dependent alterations in ILC2 and ILC3 populations and demonstrate that LSD1 repression of epithelial Cxcl16 may have a role to play in tuning recruitment of ILC2, but not ILC3 populations.

The study combines complimentary and up-to-date methods to address an important nexus of microbiota-epithelial-immune interactions in early life, and thereby seeks to define a causal role for LSD1 in regulating these interactions. While the research presented is certainly of interest, there are multiple elements that require clarification and/or further development in order to support the authors primary conclusions.

Main points:

1. As the authors state – they have previously demonstrated secretory cell deficiencies in a similar model with incomplete cKO of LSD1. Consequently, it can be argued that the same phenotypic descriptions in this study, utilising a complete cKO model, are not entirely novel. However, one advantage of the complete cKO is that it can be used to accurately address the impact of LSD1 deficiency along different regions of the GI tract. While this is partially addressed in the current submission (Fig1A-C; S1A-B) the fact that the authors have generated Swiss roll sections means that they could and should generate more granular data illustrating secretory cell alterations along both the small and large intestinal proximal-distal axis. Such granularity is important, as many elements of the murine GI tract (e.g. jejunum-ileum; mid colon-distal colon) represent a gradient on both the functional and tissue compositional levels.
2. The study relies on histological staining and mRNA expression of classical epithelial cell type marker genes (e.g. Lyz1 and Muc2) in order to infer altered maturation or abundance of these cells; however it is not clear if these cells are truly absent or rather altered. For example, Lyz1 expression is lost but what

about other Paneth cell-specific antimicrobials, or factors involved in maintaining the stem cell niche? Muc2 expression is reduced, but what about expression of other gel-forming mucins or critical mucus components? Given that the transcriptional profiles of Paneth, goblet and mature absorptive cells are now quite well established, and that the complete cKO model is free of contamination by LSD1 WT epithelial cells, it would be useful and important to provide a fuller picture of the impact of LSD1 cKO on expression of all genes associated with different epithelial cell populations.

3. Microbiota analysis highlights that impact of LSD1 cKO on post-weaning microbiota development; however, the claim that microbiota maintenance is also dependent on LSD1 is less well supported. Here the authors rely on an inducible Cre model to generate LSD1 cKO in adult mice (icKO), but the microbiota alterations observed in icKO mice do not match well with the observations in cKO mice (e.g. no consistent or strong alterations in the ratio of Bacteroidaceae:Muribaculaceae). Rather they seem to observe either very transient alterations (e.g. Bifidobacteria) or expansion of very minor components of the microbiota (e.g. Fusobacteriaceae) that correlate with loss of Lyz1 expression. The interpretation of this data in the manuscript should be revisited and reconsidered. One possibility that could be addressed is that the major alterations in cKO mice are driven by the diarrhoea phenotype that was not observed in icKO mice, and this could be examined using anti-diarrhoeal intervention (e.g. loperamide) in cKO mice.

4. Analysis of the microbiota, and interventions designed to disrupt it (e.g. ABX) must be accompanied by quantitative microbiota load data in order to provide a fuller picture of microbial dynamics. The authors should provide this information for cKO, icKO and ABX intervention studies.

5. Analysis of immune cell populations by scRNA-seq is used to highlight the potential impacts of LSD1 cKO and ABX on these populations; however, only one biological replicate is examined per genotype/condition, with the increased likelihood that any differences observed may be stochastic, and therefore less emphasis should be placed on such differences in the manuscript text. For example interpretation of differentially expressed genes in ILC2/3 populations between WT and cKO cells should bear in mind that the vast majority of these cells are isolated from the ABX treated mice, and it is therefore not possible to draw clear conclusions regarding altered expression patterns in untreated mice.

Other points:

1. Gene expression data is displayed as TPM with statistical comparisons between samples based on Benjamini–Hochberg adjusted p-values. The methods state that the authors use the Deseq2 analysis package for DGE analysis, but it is not clear if the statistical comparisons are based on DGE analysis with Deseq2 size factor normalisation or by direct comparison of TPM values. While TPM values are appropriate for comparing the relative expression of different genes within samples, they are not normally used for comparison of gene expression between samples as they do not correct well for RNA composition differences. This should be clarified and corrected if needed.

2. Some of the graphs in the paper should be more clearly labelled with what they represent rather than labels generated using various analytical programs – e.g. Fig4H-I, Fig. S4.

3. The epithelial cells in Fig. S1B appear to be upside-down.

4. There is occasional use of colloquial language (“vastly different”, “major players” etc) that should be made more accurately descriptive.

5. While many of the histology images are well presented, there are several multi-panel figures (e.g. Fig1A, Fig7, Fig S2, S6, S7) where you have to really zoom in to see what the authors are trying to show, these should be made clearer.

Reviewer #2 (Remarks to the Author):

In this manuscript, Diez-Sanchez et al. elucidated that the maturity of intestinal epithelium is related to the development of the intestinal immune system and is also important in maintaining intestinal homeostasis. In particular, the expression of *Lsd1* in adult intestinal epithelium represses *Cxcl16* expression via controlling the level of H3K4 methylation, and the CXCL16-CXCR8 axis control the population size of ILC2 and ILC3. This study provides the valuable information to understand the developmental processes related to intestinal epithelium maturation, colonization of commensal microbiome, and establishment of immune cell population, but there is still room for improvement quality of the manuscript.

Major comments

1. The intestinal cell number are reduced in cKO mice, however, height or structure of intestinal epithelium (crypt-villus) was not changed. In that case, has the composition of intestinal epithelial cells changed? (Mostly immature enterocytes?) What changes appeared in intestinal epithelium?

2. In Fig 1D, If the p-value for Crypt WT vs. Crypt cKO and Villus WT vs. Villus cKO is shown, does *Lyz1* have increased expression in villus WT, so the p-value is '****' compared to Villus cKO?

So, how much *Lyz1* in Paneth cells is expressed in villus WT?

Additionally, in Fig S1C, how much *Defa22* is also expressed in villus WT?

On the other hand, the expression of Lyz is only seen in the crypt in Fig 1A, G.

3. (Fig 1H and I) The role of BMP signaling is important in enterocyte maturation. However, since the expression level of BMP target genes is the same, there is no difference in responsiveness to BMP signaling between WT and cKO. So, why is enterocyte maturity reduced in Lsd1 cKO? Are there changes in any signaling pathways?

4. In the manuscript (line 163~), Adult cKO vs. WT has significant structural differences in the intestinal epithelium and diversity changes in the microbial composition. Is there evidence from the authors' data that differential microbial communities are important mediators of enteric and systemic inflammation?

5. Why does a decrease in Secretory cells (ex. Paneth cell, Goblet cell, and EEC) reduce enterocyte maturity? What are the regulatory factors that regulate enterocyte maturation, and how are the regulatory factors derived from secretory cells affect to enterocyte maturation?

6. In Fig 3D, stem cell expression is increased in Villus_cKO_abx, and is decreased only in the crypt.

Is it true that stem cell expression in ABX mice in Fig. 3D is high in the villus and not the crypt?

Similarly, In Fig 3E, stem cell expression in Lgr5 (ABX) Villus cKO is compared to Villus WT, and the p-value is written. It seems that these data do not match.

Additionally, in Fig 3E, How different is the expression of Lyz1(ABX) and Mki67(ABX) in Villus WT and Villus cKO? Is the p-value correct?

7. In ABX treated-cKO group, the decrease of mostly intestinal marker expression (Paneth cells, goblet cells, and endocrine cells) was observed. In particular, stem cell marker (Lgr5) was only decreased in ABX treated-cKO unlikely ABX untreated-cKO. Why the ABX treatment (depletion of microbiome) can affect the expression of stem cell marker in cKOs? Moreover, did the organoids in Figure 3I were generated from ABX untreated-WT and cKO mice? It is necessary to generate the ABX treated-WT and cKO mice for identify the stem cell functions. And authors should improve materials and method section about description of experimental group.

8. Why do population size of Bacteroidaceae and Sutterellaceae increase, and population size of Muribaculaceae and Rikenellaceae decrease in Lsd1 cKO mice? Is it due to secretory cell depletion, and do the same changes occur in Atoh1 cKO mice? And, what is the relationship between microbiome populations, such as Bacteroidaceae, Sutterellaceae, Muribaculaceae and Rikenellaceae, and enterocyte maturation?

9. Are the reduced levels of IgA in cKO mice described in Fig. 5 and Fig. S5 directly related to M cell maturation? Or, is there also a change in M cell maturation because maturation of the intestinal epithelium does not occur in cKO mice?

10. In the manuscript (line 347~), ILC2 and ILC3 have enhanced recruitment to the lamina propria in cKO mice. Is this result related to the diarrhea phenotype?

Minor comments

1. (line 105) need to delete the tracking.

2. Looking at the data in Fig. 1J-L, the size difference (Fig. 1J) and diarrhea (Fig. 1L) of the mice are consistent with the author's claim that they did not gain weight typically.

In the weight analysis, but WT vs. of cKO, it is necessary to check whether the P value for each section has been created. Also, it needs to explain the weight graph of gender aggregated data and statistical analysis.

3. If Lsd1 cKO induced diarrhea, author need to confirm the tight junction molecule expression.

4. In Fig 2A, B, although there are significant differences in the microbiota in Adult WT vs cKO, the authors explain that LSD1 deletion (icKO) in adulthood has little effect on the intestinal microbial composition in Fig. 2C-I and Fig. S2.

I wonder that in Fig S2, aren't some microbiota species (such as Ruminococcaceae, Acidaminococcaceae, Veillonellaceae, Selenomonadaceae, Barnesiellaceae, Coriobacteriaceae, and Fusobacteriaceae), which change significantly over time after Tamoxifen treatment. Does this microbial community have a significant impact?

5. To support the author's claim, it seems appropriate to place the organoid data (Fig. 3I, J) in Fig. 1.

6. In Fig 4E, G, is the breeding method (conventionally bred) of WT and Untreated mice the same? If true, how can you explain the differences in the Immune cell population of WT vs. Untreated?

7. (Fig 6E) Although the change in H3K4me1 is marginal, the expression level of Cxcl16 is significantly higher (~4 fold) in the intestinal epithelium of cKO mice compared to that of WT mice. Is the change in H3K4me1 major factor in regulating the expression of Cxcl16? Aren't there other regulatory mechanisms?

8. It is difficult to confirm RORyt and CD3 expression in Fig. 6A, 7A, Fig. S6, and S7. Since RORyt is one of the main data in this paper, a distinguishable fluorescent stain, or pseudo-color, is required.

9. (line 415) need to check the typo.

10. (line 1105) Fig. S4E should be written in bold font.

Reviewer #3 (Remarks to the Author):

This manuscript focuses primarily on the role of intestinal epithelial LSD1 in regulating immune cell composition. The authors concluded that intestinal epithelial specific deletion of LSD1 resulted in a deficiency of intestinal secretory epithelial cells, altered gut microbiota, and changed immune cell compositions. Through the microbiota depletion experiment with antibiotics, the authors concluded that the regulation of immune cells by intestinal epithelial LSD1 is not dependent on microbiota. They also discovered that LSD1 may regulate chemokines such as CXCL16 and control the accumulation of ILC2 in the intestines. The same group has previously demonstrated that LSD1 is involved in regulating intestinal epithelial cell differentiation or maturation. The novel part of this manuscript relates to the regulation of immune cells. Even though the observation is interesting, a number of serious concerns remain.

1 In previous research, the authors have demonstrated that LSD1 plays an important role in the differentiation of goblet cells and Paneth cells. In this study, LSD1 was shown to have greater influence on multiple secretory epithelial cells using a different Vil-cre stain mouse. There is, however, no clear understanding of the mechanism. Do different intestinal epithelial cells express LSD1 at different levels? Can that be the reason that LSD1 deficiency shows different phenotypes in different mouse strains?

2 The authors focus primarily on immune cells in the small intestine. What about the immune cells in the colon? Since the colon contains a greater number of commensals, the immune cells may be dependent on the microbiota in mice lacking LSD1.

3 Most immune cell changes are detected using immunofluorescence with intestine slides. How many sections and sights have been counted? The use of flow cytometry will be very useful in assessing the changes of immune cells infiltrating the intestine.

4 On Line 190, the authors claim that they “establish a model to reverse matured intestinal epithelium towards a neonatal state later in life”. The statement may not be accurate. The authors did not provide adequate evidence to support the conclusion that intestinal epithelium in the absence of LSD1 is identical to the epithelium in neonates.

5 The authors demonstrated in Figure 3 that LSD1's effect is independent of the microbiota by using WT and LSD1^{-/-} organoids. However, it is possible that the stem cells isolated from WT and LSD1^{-/-} mice may have already been imprinted by the microbiota. Vil-creERT2 mice should be used to knock out LSD1 in vitro to eliminate the influence of the microbiome.

6 On Line 258, "normally" should be removed, since splenomegaly and enlarged mLN may not be normal. It is quite interesting that, although LSD1 deficient mice have already developed severe diarrhea, there are still no differences in the immune cells in the mLN of these mice. Does the author have any information regarding the changes in the function of immune cells in the mLN?

7 On Line 277, IL10 shows a significant difference.

8 According to Figure 4G, the number of ILC2 and ILC3 has changed significantly following antibiotic treatment. In Figure 4H, how about the gene difference between WT and LSD1 cKO mice without ABX treatment?

9 On Line 305, there is no evidence that LSD1 plays a role in M cells. The item should be removed.

10 On line 324, CD163 is not a specific marker of macrophage population 3. It would be helpful if the authors could provide the expression levels of CD163 among different cell types.

11 It is difficult to conclude from Figure S5E that Paneth cells express chemokines ccl6 and ccl9.

12 On Line 354 and in Figure S6A, scRNA-seq indicates a significant induction of ILC2 and ILC3 in the intestine following ABX treatment. Nevertheless, in Figure S6B-D, there appears to be no significant

difference in ILC2 and ILC3 between the control and ABX groups. This implies that calculating cell numbers by immunofluorescence is not accurate.

13 In Figure 6D, please provide the flow cytometry figures as well as the gating strategy.

14 On Line 364, please provide references, particularly for IL-15 and IL-18 promoting ILC3 expansion in the mouse intestine.

15 On Line 380, "cxcr16" should be replaced with "cxcr6". Cxcr6 is also expressed by other T cells. Do they exhibit similar phenotypes as ILC2 or ILC3?

16 On Line 406, although there is no difference in composition of the fecal microbiota, there may be a change in its function, and also the fecal microbiota may not represent the mucosa-associated microbiota. Accordingly, it is necessary to downscale the conclusion about the microbiota-independent effect.

17 The word "zWhile" appears to be misspelled on line 415.

18 What are the consequences of the increased ILC2 and ILC3 in LSD1 cKO mice? How does this relate to the diarrhea phenotype?

Reviewer #1 (Remarks to the Author):

In this study, Díez-Sánchez et al explore the intestinal epithelial-intrinsic role of the epigenetic modifier LSD1 in regulating postnatal epithelial maturation in the context of the developing luminal microbiota and sub-epithelial immune cell populations. They initially validate that conditional LSD1 KO (cKO) mice have altered epithelial maturation along different regions of the GI tract, and that cKO mice display reduced post-weaning weight gain and diarrhoea. Furthermore, they identify striking divergence in post-weaning microbiota development; however, via application of antibiotic (ABX)-mediated microbiota suppression they illustrate that epithelial maturation deficiency in cKO mice is independent of the microbiota. They go on to explore the relative roles of the microbiota and LSD1 in regulating mucosal immunity by profiling lamina propria immune cell populations by single cell RNA sequencing (scRNA-seq), identifying several alterations with potential biological relevance. They finally focus on LSD1-dependent alterations in ILC2 and ILC3 populations and demonstrate that LSD1 repression of epithelial *Cxcl16* may have a role to play in tuning recruitment of ILC2, but not ILC3 populations.

The study combines complimentary and up-to-date methods to address an important nexus of microbiota-epithelial-immune interactions in early life, and thereby seeks to define a causal role for LSD1 in regulating these interactions. While the research presented is certainly of interest, there are multiple elements that require clarification and/or further development in order to support the authors primary conclusions.

Main points:

1. As the authors state – they have previously demonstrated secretory cell deficiencies in a similar model with incomplete cKO of LSD1. Consequently, it can be argued that the same phenotypic descriptions in this study, utilising a complete cKO model, are not entirely novel. However, one advantage of the complete cKO is that it can be used to accurately address the impact of LSD1 deficiency along different regions of the GI tract. While this is partially addressed in the current submission (Fig1A-C; S1A-B) the fact that the authors have generated Swiss roll sections means that they could and should generate more granular data illustrating secretory cell alterations along both the small and large intestinal proximal-distal axis. Such granularity is important, as many elements of the murine GI tract (e.g. jejunum-ileum; mid colon-distal colon) represent a gradient on both the functional and tissue compositional levels.

We appreciate the insightful comment regarding the relevance of exploring secretory cell alterations along the intestinal axis. In addition to the figures highlighted (Fig. 1A-C and S1A-B), we provide further spatial and temporal granularity in Fig. 2D. Notably, we introduce a novel element by employing the *VilCreERT2* inducible model, allowing for a time-dependent analysis of cell loss post-*Lsd1* recombination. This approach reveals distinct cell types being lost at varying time points and intestinal locations. In detail, we show the progressive changes in Paneth (*Lyz+*), Goblet (*Muc2+*) and enterocytes (*Nt5e+*, *Reg1+*) in response to LSD1 loss in three different SI sections (duodenum, jejunum and ileum) and distal and proximal colon. Moreover, all these analyses are performed in a time-dependent manner, adding to the data granularity request.

However, and in additive response to the reviewer's valuable suggestion, we provide new data on enteroendocrine and tuft cell progenitor gene expression, as well as formally providing tuft cell numbers utilizing newly performed DCLK1 antibody staining. We believe this addition enhances the

comprehensiveness of our study, and we are grateful for the opportunity to address this point. Please see updated Figs. 1 and S1C for Neurog3/Sox4 and S1E-G for tuft cell imaging and quantification, described in updated first results section at lines 116 to 118.

2. The study relies on histological staining and mRNA expression of classical epithelial cell type marker genes (e.g. *Lyz1* and *Muc2*) in order to infer altered maturation or abundance of these cells; however it is not clear if these cells are truly absent or rather altered. For example, *Lyz1* expression is lost but what about other Paneth cell-specific antimicrobials, or factors involved in maintaining the stem cell niche? *Muc2* expression is reduced, but what about expression of other gel-forming mucins or critical mucus components? Given that the transcriptional profiles of Paneth, goblet and mature absorptive cells are now quite well established, and that the complete cKO model is free of contamination by LSD1 WT epithelial cells, it would be useful and important to provide a fuller picture of the impact of LSD1 cKO on expression of all genes associated with different epithelial cell populations.

We acknowledge the importance of discerning whether Paneth and goblet cells are genuinely absent or merely altered, and apologize this was not clear in the initial submission. We have now reformulated the results section to convey how Paneth cells are absent and where is the data to support such claim (see rephrased first section of results described at lines 106 to 130).

Our data shows that Paneth cells are absent in LSD1-deficient epithelium. We assessed Paneth-cell specific gene expression by using Gene Set Enrichment Analysis (GSEA) in Fig. 1E. This Paneth cell-specific gene signature is comprised of 82 genes (see Table S1.genesets.xlsx, from *Haber et al* (Ref. 29)), which were all virtually absent in cKO epithelium. Furthermore, we provide protein and gene Lysozyme staining (Fig. 1A, S1D) in tissue, as well as *Lyz1* (Fig. 1D) and the antimicrobial defensin *Defa22* (Fig. S1C) gene expression from isolated epithelium. Additionally, we show how Paneth-cell granules (either by light microscopy or UEA-1 staining), a classical functional and structural landmark of these cells, are completely absent in both intestinal tissue (Fig. 1A and S2C) and organoids derived from cKO mice (Fig. 3I) and icKO mice (see new data Fig. S3E-H). In support, in our previous paper we performed *Wnt3 in situ* staining (Paneth cell-produced niche factor) and found that to be absent in LSD1-deficient crypts (Fig. 2A from *Zwiggelaar et al*, Ref. 15). Together, there is no evidence of expression of Paneth cell specific antimicrobials or niche factors in cKO crypts.

Goblet cells are both reduced in number and altered in cellular state. We suggest they are limited in their maturation, as their granules appear to be much smaller (see new updated Fig. 1A). In support, we find that *Muc2* mRNA expression is down ~100-fold whereas the reduction in number of cells is ~5 fold (see Fig. 1C&1D), suggesting the goblet cells have reduced *Muc2* expression per cell (i.e. reduced maturation). Similarly, canonical goblet cell marker *Cla1* is reduced ~800 fold (Fig. S1C). Addressing the concern about transcriptional profiling we substantiate our claims using a 409 gene goblet cell signature derived from the same *Haber et al* (Ref. 29) study. We have clarified these points now also in the first results section. (see rephrased first section of results lines 106 to 130).

In response to the reviewer's mention of the transcriptional profile of mature absorptive cells, we present GSEA signatures in Fig. 1E and 3D under "Enterocyte (mature)", a signature based on 484 genes expressed in mature enterocytes (*Haber et al* (Ref. 29)). Additionally, the transcriptional changes in gene clusters (C1-C5) defining enterocyte location across the villus axis and reflecting maturation/specialization status, derived from Moor et al (Ref. 30) and included in Table S1. This RNA-seq analysis demonstrated the relative upregulation of genes associated with immature enterocytes (Cluster C1) and the downregulation of genes defining a mature enterocyte signature

(C4-C5) in intestinal villi derived from both untreated (Fig. 1F) and ABX-treated cKO mice (Fig. 3F). Supporting these findings, FISH in intestinal epithelium is performed in Fig. 1G and 2D using canonical markers from each cluster. We believe these additional analyses contribute significantly to a comprehensive understanding of the impact of LSD1 cKO on the expression of genes associated with various epithelial cell populations. We have modified the text to clarify this better (lines 135-137)

3. Microbiota analysis highlights that impact of LSD1 cKO on post-weaning microbiota development; however, the claim that microbiota maintenance is also dependent on LSD1 is less well supported. Here the authors rely on an inducible Cre model to generate LSD1 cKO in adult mice (icKO), but the microbiota alterations observed in icKO mice do not match well with the observations in cKO mice (e.g. no consistent or strong alterations in the ratio of Bacteroidaceae:Muribaculaceae). Rather they seem to observe either very transient alterations (e.g. Bifidobacteria) or expansion of very minor components of the microbiota (e.g. Fusobacteriaceae) that correlate with loss of *Lyz1* expression. The interpretation of this data in the manuscript should be revisited and reconsidered. One possibility that could be addressed is that the major alterations in cKO mice are driven by the diarrhoea phenotype that was not observed in icKO mice, and this could be examined using anti-diarrhoeal intervention (e.g. loperamide) in cKO mice.

We appreciate the thoughtful feedback from the reviewer, and in short, we agree. Hence, we have revised our claims and adjusted the text accordingly, please see the modifications made to the result section title (line 177). In addition, we now highlight the likelihood of diarrhea having a strong effect on the microbiota (Lines 189-195 (WT vs cKO) and Line 230-232 (within the WT vs icKO data). Finally, we acknowledge the potential value of a loperamide intervention to further investigate the role of diarrhea in microbial shifts. However, we feel that, while intriguing for future exploration, it lies outside the current scope of this article with a focus on epithelial-immune cell interplay. Nevertheless, we hope these modifications successfully address the concerns raised by the reviewer and enhance the overall clarity and validity of our findings.

4. Analysis of the microbiota, and interventions designed to disrupt it (e.g. ABX) must be accompanied by quantitative microbiota load data in order to provide a fuller picture of microbial dynamics. The authors should provide this information for cKO, icKO and ABX intervention studies.

We appreciate the suggestion to include quantitative microbiota load data, we now have performed new experiments to determine the relative bacterial load in stool samples by qPCR targeting the 16S rRNA gene. Most important for our story, we confirm a near complete loss of bacterial load upon ABX treatment (Ct values of those samples were 1 or 2 Ct values from the water control, so towards undetectable). These new results have been incorporated in Fig. S3A, described in line 240, and we have included a dedicated methods section.

Unfortunately, we did not have additional stool samples from the icKO experiment, as all was used for the sequencing experiment. Importantly for our story, from a week after recombination onwards, when the microbiota composition is nearly identical and no diarrhea is yet observed, we already confirm the changes in ILCs. Therefore, the main storyline is not affected by the lack of stool samples to measure bacterial load in the icKO experiment.

5. Analysis of immune cell populations by scRNA-seq is used to highlight the potential impacts of LSD1 cKO and ABX on these populations; however, only one biological replicate is examined per

genotype/condition, with the increased likelihood that any differences observed may be stochastic, and therefore less emphasis should be placed on such differences in the manuscript text. For example interpretation of differentially expressed genes in ILC2/3 populations between WT and cKO cells should bear in mind that the vast majority of these cells are isolated from the ABX treated mice, and it is therefore not possible to draw clear conclusions regarding altered expression patterns in untreated mice.

In principle, we agree with the reviewer that an n=1 is insufficient to draw conclusions, especially on something so stochastic as gene expression. Indeed, our aim was to use the differential gene expression data of Fig. 6H&I to guide us towards the CXCL16 experiment of Fig. 7. Nonetheless, we have made additional notes highlighting this is n=1, and that this limits the conclusions. (See Lines 414 and 425-426)

Other points:

1. Gene expression data is displayed as TPM with statistical comparisons between samples based on Benjamini–Hochberg adjusted p-values. The methods state that the authors use the Deseq2 analysis package for DGE analysis, but it is not clear if the statistical comparisons are based on DGE analysis with Deseq2 size factor normalisation or by direct comparison of TPM values. While TPM values are appropriate for comparing the relative expression of different genes within samples, they are not normally used for comparison of gene expression between samples as they do not correct well for RNA composition differences. This should be clarified and corrected if needed.

We apologize for adding unclear figure labels. The p-values in the graphs using TPM values are Benjamini-Hochberg adjusted p-values calculated with DESeq2 and therefore use the size normalization of DESeq2 for the statistics. We have updated all relevant figure legends to clearly state that the p-values are from DESeq2. Here is one example of change from Fig. 1D:

*“(Benjamini-Hochberg adjusted p-value **calculated with DESeq2**)”*

TPM values are not a perfect normalization and could be influenced by differences in RNA composition. However, it is the best unit out of the common units for presenting batch RNAseq data as it normalizes for both gene length and sequencing depth. Instead of displaying TPM values we could present the data using the log₂(foldchange) estimated by DESeq2. However, TPM values gives information about the variation in replicates in a way that the aggregated log₂(foldchange) value from DESeq2 would not. We therefore think displaying TPM values is the best way to present these data. TPM is also commonly used to present expression data, gene expression data in the protein atlas is for example presented using a version of this unit (<https://www.proteinatlas.org/ENSG00000090382-LYZ/tissue>).

2. Some of the graphs in the paper should be more clearly labelled with what they represent rather than labels generated using various analytical programs – e.g. Fig4H-I, Fig. S4.

Apologies for the oversight, the graph labelling has been corrected in both figures in the latest version and legends updated for clarification.

3. The epithelial cells in Fig. S1B appear to be upside-down.

Thanks for pointing that out, Fig. S1B has been corrected. The apical-basal axis is now displayed in the correct direction.

4. There is occasional use of colloquial language (“vastly different”, “major players” etc) that should be made more accurately descriptive.

We appreciate the constructive feedback provided by the reviewer. In response to the observation about occasional colloquial language, we have carefully reviewed the manuscript and made the necessary modifications to ensure a more accurate and descriptive language throughout.

5. While many of the histology images are well presented, there are several multi-panel figures (e.g. Fig1A, Fig7, Fig S2, S6, S7) where you have to really zoom in to see what the authors are trying to show, these should be made clearer.

Thank you for bringing attention to this matter. In response to the concern about the clarity of certain histology images, we have implemented the following changes to enhance interpretability:

- Fig. 1A: While intentionally presented with a zoomed-out view to illustrate tissue-wide recombination efficiency, an inset has been added to focus on an individual villus. This modification provides a detailed representation of the loss of Paneth cells and the decrease and lack of maturation in goblet cells.
- Fig. S2: To improve readability of fluorescent stains, Fig. S2 has been separated into two pages, allowing Fig. S2B to take the entire page width. Additionally, an inset is included to enhance visibility.
- Fig. 6A: The figure now spans the entirety of the page width, contributing to improved visualization.
- Fig. S6D: The figure now spans the entirety of the page width, contributing to improved visualization. Fig. S6 now spans two pages of supplementary material.
- Fig. 7A: We have adjusted Fig. 7A to showcase a single zoomed-in villus for easier interpretation. A corresponding supplementary figure is now available, presenting a broader field of view for representativeness.
- Fig. S7A: Magnification has been applied to Fig. S7A for better visualization.

It's important to note that for fluorescent microscopy panels featuring only one villus or merged channels, corresponding supplementary figures have been included. These supplementary figures either show a larger field of view or present individual channels separately to facilitate interpretation.

We value the feedback provided by the reviewer, and we believe that these changes have significantly improved the readability and interpretability of the histology images in question.

Reviewer #2 (Remarks to the Author):

In this manuscript, Diez-Sanchez et al. elucidated that the maturity of intestinal epithelium is related to the development of the intestinal immune system and is also important in maintaining intestinal homeostasis. In particular, the expression of Lsd1 in adult intestinal epithelium represses Cxcl16 expression via controlling the level of H3K4 methylation, and the CXCL16-CXCR8 axis control the population size of ILC2 and ILC3. This study provides the valuable information to understand the developmental processes related to intestinal epithelium maturation, colonization of commensal microbiome, and establishment of immune cell population, but there is still room for improvement quality of the manuscript.

Major comments:

1. The intestinal cell number are reduced in cKO mice, however, height or structure of intestinal epithelium (crypt-villus) was not changed. In that case, has the composition of intestinal epithelial cells changed? (Mostly immature enterocytes?) What changes appeared in intestinal epithelium?

Apologies for the misunderstanding; there are no fewer numbers of intestinal epithelial cells per crypt-villus unit. Therefore, as pointed out by the reviewer, there are no structural differences in the epithelium. However, epithelial cellular type composition is drastically different. For instance, there is a complete loss of Paneth cells (see Fig. 1A and 1B) and reduced numbers of goblet cells (see Fig. 1A and 1C). Furthermore, as highlighted by the reviewer, we observe a relative expansion of immature enterocytes at the expense of mature ones. An example can be seen in Fig. 1F, where we compare complete villi gene expression to spatially restricted gene signatures derived from Ref. 30. There is an upregulation of genes associated with clusters C1 and C2, which are linked to less differentiated enterocytes located at the bottom of the villi. This is further supported by the extension of the C1 marker, Reg1, along the entirety of the villi and the loss of Nt5e, a C5 marker for mature enterocytes/tip villi, as illustrated in Fig. 1F.

We hope this explanation provides a satisfactory clarification of the changes observed in the composition of intestinal epithelial cells in the cKO mice. We have thoroughly rephrased the first results section to provide more clarity.

2. In Fig 1D, If the p-value for Crypt WT vs. Crypt cKO and Villus WT vs. Villus cKO is shown, does Lyz1 have increased expression in villus WT, so the p-value is '****' compared to Villus cKO? So, how much Lyz1 in Paneth cells is expressed in villus WT? Additionally, in Fig S1C, how much Defa22 is also expressed in villus WT? On the other hand, the expression of Lyz is only seen in the crypt in Fig 1A, G.

Thank you for bringing up this comment; it has prompted us to modify our data presentation for RNA-seq in Figs. 1D, S1C, 3E and S3A to prevent potential misunderstandings. The way we depict the data for crypts and villi aims to highlight the precision of our tissue processing for RNA-seq. Markers like Lyz1/ Defa22, specific to Paneth cells, and Lgr5 / Olfm4, associated with stem cells, show a nearly exclusive expression in the crypts, aligning with their cellular localization. However, we could perform statistical analysis on these samples (the p values initially noted, and this is indeed comparing villus WT to villus cKO).

For certain genes considered canonical markers of different cell types in intestinal crypts, any expression in the villi compartment is either a result of minimal cross-contamination during physical separation or represents basal expression present in all epithelial cells. Therefore, the statistical and biological significance of these particular markers applies specifically to the crypt compartment. The expression data in the villi serve as a technical control, illustrating their near absence and lack of biological significance in the comparison between WT and cKO.

To address this, we have removed statistical significance from the villus compartments for markers like *Lgr5*, *Olfm4*, *Lyz*, and *Defa22*, marking them as "n/a" (not applicable) to avoid potential misinterpretations and updated the corresponding Statistical Analysis section under Materials & Methods.

3. (Fig 1H and I) The role of BMP signaling is important in enterocyte maturation. However, since the expression level of BMP target genes is the same, there is no difference in responsiveness to BMP signaling between WT and cKO. So, why is enterocyte maturity reduced in *Lsd1* cKO? Are there changes in any signaling pathways?

This is a great question. Simply, we do not know. Our current hypothesis is that the epigenetic state of KO cells is restricting functional maturation of enterocytes (and all lineages really). In other words, even though receptors and initial target gene expression (such as we show) is working, the eventual cellular programming needed for maturation, which requires massive transcriptomic changes is not happening. We're now trying to design experiments that would test this hypothesis, but unfortunately that will take several years, and will be a new story.

4. In the manuscript (line 163~), Adult cKO vs. WT has significant structural differences in the intestinal epithelium and diversity changes in the microbial composition. Is there evidence from the authors' data that differential microbial communities are important mediators of enteric and systemic inflammation?

While it is a possibility we considered, as indicated by the experiments presented in Figure 4 and S4, our data does not reveal differences in spleen or mLN to suggest systemic inflammation. Specifically, no discernible changes in immune cell composition—comprising T cells, B cells, and antigen-presenting cells—were observed in either spleen (Fig. S4A) or mLN-derived CD45+ cells (Fig. S4B), as evaluated through flow cytometry. This also included the numbers of CD4+ CD25+ regulatory T (Treg) cells (Fig. S4A & S4B) – a hallmark cell type that is expected to be altered during systemic inflammation.

Lastly, in our scRNAseq study, aligning with our observation that cKO mice do not develop overt inflammation, we did not detect significant alterations in the expression of inflammation-associated cytokines (*Ifng*, *Il1b*, *Tnf*, *Il22* and *Il17a*) in immune cells derived from cKO animals compared to those from WT animals (Fig. S4D).

5. Why does a decrease in Secretory cells (ex. Paneth cell, Goblet cell, and EEC) reduce enterocyte maturity? What are the regulatory factors that regulate enterocyte maturation, and how are the regulatory factors derived from secretory cells affect to enterocyte maturation?

The relationship between secretory cell populations and enterocyte maturation dogmatically follows a binary pattern, where an increase in one often coincides with a decrease in the other. This aligns with the previous question regarding why enterocytes fail to mature.

Regarding the crosstalk between the regulatory factors secreted by these cells, examining the data presented in Fig. 2D, illustrating the sequential loss of secretory cells like Paneth and Goblet cells, sheds light on this relationship. Specifically, the expression pattern of Reg1, a marker for bottom villi enterocytes associated with a less differentiated state, extends along the length of the villi one week post TAM-induced Lsd1 recombination. This timeframe coincides with a period when Paneth and Goblet cells are still present, as indicated by the Lyz1 and Muc2 markers. Hence, based on our data, enterocyte maturation does not seem to be strongly affected by the presence or absence of these secretory cells.

6. In Fig 3D, stem cell expression is increased in Villus_cKO_abx, and is decreased only in the crypt. Is it true that stem cell expression in ABX mice in Fig. 3D is high in the villus and not the crypt? Similarly, In Fig 3E, stem cell expression in Lgr5 (ABX) Villus cKO is compared to Villus WT, and the p-value is written. It seems that these data do not match.

Apologies for the confusion created and thanks for pointing out this issue. To enhance heatmap readability and avoid data misinterpretation, we have revised the legends and figures (Fig. 1E, 1I, 3D and 3H) to clarify that the data representation originates from the cKO vs WT gene expression comparison across different tissue compartments and treatment conditions (i.e. crypt_cKO should be read as crypt_cKO_vs_WT). Please also note that grey color indicates that the GSEA is not significant, and we therefore cannot conclude about direction of enrichment. This is for example the case with crypt_cKO vs WT (ABX) in Fig. 3D.

Additionally, in Fig 3E, How different is the expression of Lyz1(ABX) and Mki67(ABX) in Villus WT and Villus cKO? Is the p-value correct?

This is related to point 2 raised by the reviewer. In this case, both Lyz1 and Mki67 are crypt markers, so we have removed p values for the villus data and put n/a (not applicable in Fig. 3E), so there's no misinterpretation of the data.

7. In ABX treated-cKO group, the decrease of mostly intestinal marker expression (Paneth cells, goblet cells, and endocrine cells) was observed. In particular, stem cell marker (Lgr5) was only decreased in ABX treated-cKO unlikely ABX untreated-cKO. Why the ABX treatment (depletion of microbiome) can affect the expression of stem cell marker in cKOs? Moreover, did the organoids in Figure 3I were generated from ABX untreated-WT and cKO mice? It is necessary to generate the ABX treated-WT and cKO mice for identify the stem cell functions. And authors should improve materials and method section about description of experimental group.

We thank the reviewer for this comment. We'd like to add the following:

- ISC signature (by GSEA) is modestly enriched in cKOs vs WT, but *Lgr5* itself is unchanged (Fig. 1D&E)
- ISC signature (by GSEA) is unchanged in ABX cKOs vs WT, but *Lgr5* itself is modestly reduced (Fig. 3D&E)
- ISC signature by GSEA and *Lgr5* expression from WT vs cKO organoids are both unchanged (Fig. 3D&3J).

Our rationale for not (yet) pursuing this in detail is because this is data from crypt or organoid bulk RNA-seq, where the relative ISC gene expression will be strongly modulated by the presence and absence of other cell types, including notably the Paneth cells in WT compared to cKO (that don't have Paneth cells). Together, our conclusion is that there are no consistent, or more importantly, strong changes to the relative ISC population. In addition, we now provide new data by deleting LSD1 *in vitro* to test long-term microbial imprinting in ISCs, and essentially phenocopy cKO organoids (and tissue epithelium), thereby more formally ruling out a role of the microbiota on ISC biology within the context of LSD1 deletion. (New Fig. S3D-H, lines 269-276). In addition, we have improved the Material and methods section describing the experimental groups and revised the corresponding figure legends.

8. Why do population size of Bacteroidaceae and Sutterellaceae increase, and population size of Muribaculaceae and Rikenellaceae decrease in Lsd1 cKO mice? Is it due to secretory cell depletion, and do the same changes occur in Atoh1 cKO mice? And, what is the relationship between microbiome populations, such as Bacteroidaceae, Sutterellaceae, Muribaculaceae and Rikenellaceae, and enterocyte maturation?

We thank the reviewer for bringing in these intriguing aspects. Our primary aim was to characterize microbial populations, initially to identify differences between WT and cKO mice, and subsequently to eliminate them using antibiotic treatment, demonstrating the independent impact of intestinal epithelium maturation on certain immune cells regardless of microbiome alterations.

Regarding the shifts in bacterial populations, while we could provide some insights based on the temporal changes in certain bacterial species and the concurrent loss of secretory cells (based on Fig. 2 and Fig. S2), the multitude of influencing factors complicates a definitive interpretation. Notably, the diarrhea phenotype in cKO mice might contribute to observed changes in Bacteroidaceae and Muribaculaceae, as discussed in a related study (new Ref. 34):

We have incorporated this aspect into the manuscript see lines 188-194, 228-232 and 484-495.

Concerning Atoh1 mice, unfortunately, we haven't found literature addressing the same changes.

As for the relationship between bacterial species and enterocyte maturation within the context of our study, our data suggest that LSD1-driven epithelial maturation occurs independently of the microbiota (Fig. 3 and Fig. S3). A recent study (<https://www.ncbi.nlm.nih.gov/pmc/articles/PMC8346670/>) has shown that overall, antibiotic use leads to a slight increase of enterocyte maturation, while reducing levels of antimicrobials (such as those expressed by Paneth cells). However, we are not aware of any studies specifically linking enterocyte maturation with specific microbiota populations.

9. Are the reduced levels of IgA in cKO mice described in Fig. 5 and Fig. S5 directly related to M cell maturation? Or, is there also a change in M cell maturation because maturation of the intestinal epithelium does not occur in cKO mice?

Unfortunately, we did not seem to have Peyer's patches within most of our histology samples, so we cannot address this in detail. However, we do think that the lack of goblet cells (goblet cells also passage antigen, especially early in life (Refs. 23 and 24) and potentially reduced M cell maturation is causing the reduced IgA levels. We speculate on this in manuscript lines 334-337

10. In the manuscript (line 347~), ILC2 and ILC3 have enhanced recruitment to the lamina propria in cKO mice. Is this result related to the diarrhea phenotype?

The enhanced recruitment of ILC2 and ILC3 to the lamina propria in cKO mice is not likely related to the diarrhea phenotype. This conclusion is supported by the observation of a similar recruitment pattern in icKO mice, where the increase in ILCs happens before the onset of diarrhea (See Fig. 7) and described in lines 440-441).

Minor comments

1. (line 105) need to delete the tracking.

We thank the reviewer for this finding, and we have corrected the mistake.

2. Looking at the data in Fig. 1J-L, the size difference (Fig. 1J) and diarrhea (Fig. 1L) of the mice are consistent with the author's claim that they did not gain weight typically.

In the weight analysis, but WT vs. of cKO, it is necessary to check whether the P value for each section has been created. Also, it needs to explain the weight graph of gender aggregated data and statistical analysis.

The requested information is provided in the legend for Fig. 1K: "Gender-aggregated weights are presented as mean \pm SEM; n = 12 mice (6 male & 6 female)/genotype/timepoint from 4 independent experiments (Individual timepoints assessed by two-tailed unpaired t-test)." The p-value for each section has been calculated, and the figure illustrates timepoints with the same significance aggregated and separated by discontinuous lines to avoid crowding the figure. This clarification has now been incorporated into the figure legend.

3. If Lsd1 cKO induced diarrhea, author need to confirm the tight junction molecule expression .

In response to the reviewer's suggestion, we conducted ZO-1 (tight junction) and E-cadherin (adherens junction) stainings to assess the junctional status across the intestinal epithelium in WT and cKO mice. Unfortunately, the 2 ZO-1 antibodies we tried did not work. So, we only provide E-cadherin staining that is now included (See new Fig S1H, lines 167-171). We didn't observe clear changes, which fits with a lack of local and systemic inflammatory profile. Hence the diarrhea we observe in cKO mice can be osmotic diarrhea caused by malabsorption, which fits with the changes in the microbiota.

4. In Fig 2A, B, although there are significant differences in the microbiota in Adult WT vs cKO, the authors explain that LSD1 deletion (icKO) in adulthood has little effect on the intestinal microbial composition in Fig. 2C-I and Fig. S2.

I wonder that in Fig S2, aren't some microbiota species (such as Ruminococcaceae, Acidaminococcaceae, Veillonellaceae, Selenomonadaceae, Barnesiellaceae, Coriobacteriaceae, and Fusobacteriaceae), which change significantly over time after Tamoxifen treatment. Does this microbial community have a significant impact?

We agree these species could have certain niche-specific impacts, which in part was our reasoning for providing these detailed graphs (so that the community can relate it to their work relating to goblet / Paneth cell changes). However, because our storyline is not focusing on this, we feel additional descriptions would dilute the manuscript too much.

5. To support the author's claim, it seems appropriate to place the organoid data (Fig. 3I, J) in Fig. 1.

While we appreciate the suggestion, we intentionally placed the organoid data (Fig. 3I, J) in Fig. 3 to emphasize the specific focus of our study. Our objective is to demonstrate the isolated impact of LSD1 on intestinal epithelial maturation, independent of external factors. Placing the *in vitro* organoid data in Fig. 3 aligns with our narrative, highlighting that LSD1 directs the intestinal epithelial maturation process intrinsically, independently of external factors such as the microbiota (ABX-derived data) or the tissue niche (organoid-derived data). In this line, and as part of Fig. S3, we have included a new experiment where we induce LSD1 recombination *in vitro* (Fig. S3D-H). This last experiment provides confirmation of the microbiota-independent role of LSD1 in epithelial maturation.

6. In Fig 4E, G, is the breeding method (conventionally bred) of WT and Untreated mice the same? If true, how can you explain the differences in the Immune cell population of WT vs. Untreated?

Apologies for the confusion, there are 4 types of mice here WT, cKO, WT (ABX), cKO (ABX).

In Fig. 4E we pooled WT+WT(ABX) and we pooled cKO+cKO(ABX) samples –to highlight the changes caused by genotype (WT vs cKO) irrespective of ABX and hence of microbiota.

In Fig. 4G we pooled WT+cKO and we pooled WT(ABX)+cKO(ABX) samples –to assess the changes caused by treatment ('untreated' vs ABX) irrespective of genotype and hence of epithelial maturation.

So, 'WT' and 'Untreated' are 2 different samples pooled as described above. We hope this has clarified the issue. For clarity, we have altered the text describing this to avoid misunderstandings. (See lines 308-310).

7. (Fig 6E) Although the change in H3K4me1 is marginal, the expression level of Cxcl16 is significantly higher (~4 fold) in the intestinal epithelium of cKO mice compared to that of WT mice. Is the change in H3K4me1 major factor in regulating the expression of Cxcl16? Aren't there other regulatory mechanisms?

As indicated in the manuscript, our approach involved leveraging the methylation profile of Lsd1 KO mice from a previous study to identify potential cytokines interacting with receptors on innate lymphoid cells (ILC2s and ILC3s). Note that this data has derived from an older Villin-Cre Lsd1 fl/fl strain where 10-30% of the tissue was unrecombined and still expressed Lsd1, and thus the change in methylation is an underrepresentation. Cxcl16 emerged due to its distinct methylation profile in cKO, and its upregulation (approximately 4-fold) was considered a candidate for interaction with the CXCR6 receptor in these cells.

However, we acknowledge the reviewer's valid point. In response, we have tempered our claims (see line 406) in the manuscript, acknowledging the possibility of additional regulatory mechanisms influencing Cxcl16 expression beyond chromatin structure remodeling in response to the methylation profile. We appreciate the reviewer's insightful feedback, and Fig. 6E has been adjusted to better illustrate differences in the methylation profile surrounding the Cxcl16 gene. Thank you for your valuable input.

8. It is difficult to confirm RORyt and CD3 expression in Fig. 6A, 7A, Fig. S6, and S7. Since RORyt is one of the main data in this paper, a distinguishable fluorescent stain, or pseudo-color, is required.

Thank you for bringing our attention to this matter. In response to the concern about the clarity of certain histology images, we have implemented the following changes to enhance interpretability:

- Fig. 1A: While intentionally presented with a zoomed-out view to illustrate tissue-wide recombination efficiency, an inset has been added to focus on an individual villus. This modification provides a detailed representation of the loss of Paneth cells and the decrease in goblet cells.
- Fig. S2: To improve readability of fluorescent stains, Fig. S2 has been separated into two pages, allowing Fig. S2B to take the entire page width. Additionally, an inset is included to enhance visibility.
- Fig. 6A: The figure now spans the entirety of the page width, contributing to improved visualization.
- Fig. S6D: The figure now spans the entirety of the page width, contributing to improved visualization. Fig. S6 now spans two pages of supplementary material.
- Fig. 7A: We have adjusted Fig. 7A to showcase a single zoomed-in villus for easier interpretation. A corresponding supplementary figure is now available, presenting a broader field of view for representativeness.
- Fig. S7A: Magnification has been applied to Fig. S7A for better visualization.

It's important to note that for fluorescent microscopy panels featuring only one villus or merged channels, corresponding supplementary figures have been included. These supplementary figures either show a larger field of view or present individual channels separately to facilitate interpretation as requested.

We value the feedback provided by the reviewer, and we believe that these changes have significantly improved the readability and interpretability of the histology images in question.

9. (line 415) need to check the typo.

Apologies, it is now fixed.

10. (line 1105) Fig. S4E should be written in bold font.

We consistently refer to other figures without using bold font in the figure legend text (unlike in the main text); we will adhere to the journal guidelines during final copy editing.

Reviewer #3 (Remarks to the Author):

This manuscript focuses primarily on the role of intestinal epithelial LSD1 in regulating immune cell composition. The authors concluded that intestinal epithelial specific deletion of LSD1 resulted in a deficiency of intestinal secretory epithelial cells, altered gut microbiota, and changed immune cell compositions. Through the microbiota depletion experiment with antibiotics, the authors concluded that the regulation of immune cells by intestinal epithelial LSD1 is not dependent on microbiota. They also discovered that LSD1 may regulate chemokines such as CXCL16 and control the accumulation of ILC2 in the intestines. The same group has previously demonstrated that LSD1 is involved in regulating intestinal epithelial cell differentiation or maturation. The novel part of this manuscript relates to the regulation of immune cells. Even though the observation is interesting, a number of serious concerns remain.

1. In previous research, the authors have demonstrated that LSD1 plays an important role in the differentiation of goblet cells and Paneth cells. In this study, LSD1 was shown to have greater influence on multiple secretory epithelial cells using a different Vil-cre stain mouse. There is, however, no clear understanding of the mechanism. Do different intestinal epithelial cells express LSD1 at different levels? Can that be the reason that LSD1 deficiency shows different phenotypes in different mouse strains?

We think the full cKO (this study) has a stronger more profound (but the same) phenotype as the previous cKO (previous work). We do not yet fully understand the mechanism, but with regard to other secretory cells (i.e. enteroendocrine (EE)), we observe an increase of their immature gene profile (e.g. *Neurog3* expression is 5-10 fold increased in cKO crypts), but a lack of maturation results in a much reduced expression of a mature EE marker such as *ChgA* (Fig. 1D) or the whole gene set by GSEA (Fig. 1E). Thus, we think there is an overall lack of maturation, in addition to the differentiation phenotype where reduced PC/GC leads to increase in EE progenitors. Of note, LSD1 is differentially expressed in the epithelium, with higher levels in the crypt (although nearly absent in Paneth cells), but as mentioned above, we do not think there is a different phenotype compared to previous work. We have clarified this in a now updated first section of the results and have included the new *Neurog3* (and *Sox4*) expression data (new Fig. S3C, and lines 116-118)

2. The authors focus primarily on immune cells in the small intestine. What about the immune cells in the colon? Since the colon contains a greater number of commensals, the immune cells may be dependent on the microbiota in mice lacking LSD1.

We agree this would be interesting to test, especially with regard to the microbiota. However, we feel this is beyond the scope of the current study to also assess the colon immune cell composition in detail.

3. Most immune cell changes are detected using immunofluorescence with intestine slides. How many sections and sights have been counted?

For each biological replicate, we collected a representative 10cm segment of the intestine, encompassing the duodenum, jejunum, and ileum, and processed it into a spiraled swiss-roll. To quantify cells, we randomly selected complete and intact villi across the entire swiss roll. Approximately 25-30 morphologically complete villi were counted and averaged per biological replicate. This approach ensured robust data collection while addressing the challenges posed by the

nature of the samples (incomplete or rotated villi per section, presence of cryptopatches). We have now included this information under Image Acquisition, Processing Software and Quantification paragraph in the methods section.

The use of flow cytometry will be very useful in assessing the changes of immune cells infiltrating the intestine.

While we acknowledge the potential advantages of flow cytometry in immune cell population analysis, several practical considerations led us to favor confocal microscopy in the majority of our study. Lamina propria leukocyte isolation is a time-intensive process with constraints on the number of mice that can be processed simultaneously due to rapid viability decline (and it is likely that some cell types are more susceptible to viability issues than other, which would skew the results). Given that our experimental design typically involves at least four mice per group/condition/time point, processing all samples at once becomes challenging. Confocal microscopy, however, offers distinct advantages. It allows us to maintain the spatial resolution necessary for quantifying specific cell populations, particularly the ILCs within the villi, as well as the macrophage population underneath the crypts (Fig. 5). Moreover, it enables us to avoid potential contamination from Peyer's patches and cryptopatches, which could significantly impact immune cell proportions in flow cytometry if not carefully controlled.

4. On Line 190, the authors claim that they “establish a model to reverse matured intestinal epithelium towards a neonatal state later in life”. The statement may not be accurate. The authors did not provide adequate evidence to support the conclusion that intestinal epithelium in the absence of LSD1 is identical to the epithelium in neonates.

We appreciate the reviewer's insight, and we acknowledge that our evidence may not fully substantiate the conclusion that the intestinal epithelium in the absence of LSD1 is entirely identical to neonatal epithelium. We have now modified the statement and now accurately conveys the essence of our study without overemphasizing the one-to-one equivalence to neonatal epithelium. The revised text (lines 212-213) reads as follows: “We hereby establish a model to reverse matured intestinal epithelium towards a neonatal state status that recapitulates features of neonatal/developing epithelium”.

5. The authors demonstrated in Figure 3 that LSD1's effect is independent of the microbiota by using WT and LSD1^{-/-} organoids. However, it is possible that the stem cells isolated from WT and LSD1^{-/-} mice may have already been imprinted by the microbiota. Vil-creERT2 mice should be used to knock out LSD1 *in vitro* to eliminate the influence of the microbiome.

We appreciate the insightful suggestion by the reviewer. To address the concern regarding potential microbiota imprinting on stem cells, we performed new additional experiments the reviewer suggested using Vil-CreERT2 derived organoids to knock out LSD1 *in vitro*. The results, now included in Fig. S3, mirror the findings from Figure 3, confirming the microbiota-independent effect of LSD1 on organoids treated with tamoxifen after crypt isolation (and hence independently of differential microbial imprinting).

6. On Line 258, "normally" should be removed, since splenomegaly and enlarged mLN may not be normal. It is quite interesting that, although LSD1 deficient mice have already developed severe

diarrhea, there are still no differences in the immune cells in the mLN of these mice. Does the author have any information regarding the changes in the function of immune cells in the mLN?

We appreciate the reviewer's observation, and we have removed the term 'normally' as suggested. In our investigation of LSD1-deficient mice, we did examine the total number of isolated cells from mLN (Fig. 4C) and characterized the major immune cell populations through flow cytometry (Fig. S4B). However, our focus was on establishing the broad immune cell composition. While we acknowledge that changes in the function of immune cells can be crucial, our experimental scope was oriented toward providing a comprehensive overview of immune cell composition in the context of LSD1 deficiency. Given the breadth of our analyses, we currently do not possess specific information regarding alterations in the functional aspects of mLN-derived cells.

Related to the diarrhea phenotype and after providing new data on the integrity of the epithelial barrier (see new Fig. S1H, described 167-171) we conclude barrier remains unaffected. Instead, we now provide new data suggesting these mice may suffer from malabsorptive diarrhea, which is probably related to the lack of genes important for nutrient absorption (See new Figs. S2B & S3B, described in lines 193-195 and 243-245).

7. On Line 277, IL10 shows a significant difference.

We have made amendments to the results section to highlight that, as indicated by the scRNAseq data, IL10 exhibits variations in the number of expressing cells, especially within the myeloid cluster.

8. According to Figure 4G, the number of ILC2 and ILC3 has changed significantly following antibiotic treatment. In Figure 4H, how about the gene difference between WT and LSD1 cKO mice without ABX treatment?

The apparent increase in the number of ILC2s and ILC3s following antibiotic treatment is influenced by a reduction in the myeloid cell ratio within the CD45+ population (as expected under microbiota depleting conditions, <https://doi.org/10.3389/fphys.2018.01534>).

In untreated mice, the myeloid population dominates the single-cell sequencing, leaving limited slots for the detection of less abundant populations such as ILC2s, ILC3s, and T cells. As observed in Fig. 6A, 6B, and S6B, the total number of ILC2s and ILC3s remains largely unchanged upon ABX treatment. This emphasizes the strength of microscopy over flow cytometry in detecting low-frequency populations during significant shifts in the overall CD45+ cell composition (related to the reviewer's point 3).

To address the second question, due to the low number of sequenced ILC2s and ILC3s in untreated mice, it becomes challenging to compare the number of significantly changed genes (n_{sign}) with DC, pDCs, and plasma cells. However, the comparison of n_{sign} between cKO and WT in untreated mice reveals: DC (44), ILC-2 (0), ILC-3 (5), pDC (1), and plasma cells (77).

Rather than as a quantitative measure, the UMAPs shown in Fig. 6E, 6F and 6G, are depicted to illustrate the clustering distribution of the immune cells across genotypes and treatments. The rationale for this approach is described in lines 293-298 of the manuscript.

9. On Line 305, there is no evidence that LSD1 plays a role in M cells. The item should be removed.

Agreed, we have removed the data on M cells.

10. On line 324, CD163 is not a specific marker of macrophage population 3. It would be helpful if the authors could provide the expression levels of CD163 among different cell types.

We appreciate the reviewer's observation, the text has been revised for accuracy. We have modified so it describes the fact that this subcluster M ϕ 3 co-expresses Cd163 and F13a1, recently described markers "specific" for a macrophage subset residing underneath intestinal crypts.

Regarding expression levels of Cd163 among different cell types (as now shown in Fig. S5A) we have also plotted Cd163 and F13a1 expression levels across myeloid subclusters as per reviewer's request. Cd163 and F13a1 expression is mainly localized within subcluster M ϕ 3 as shown in the violin plots below (we unfortunately do not have a working antibody against F13a1).

11. It is difficult to conclude from Figure S5E that Paneth cells express chemokines ccl6 and ccl9.

The manuscript has been updated (lines 369-372) to enhance the accuracy of the expression pattern description, and we also refer to a paper showing high expression of CCL6 at the bottom of crypts (Ref. 48).

To address potential confusion related to the color coding of Enterocyte (Progenitor) and Paneth cell clusters, we have revised Fig. S5D (previously S5E) to improve readability.

12. On Line 354 and in Figure S6A, scRNA-seq indicates a significant induction of ILC2 and ILC3 in the intestine following ABX treatment. Nevertheless, in Figure S6B-D, there appears to be no significant difference in ILC2 and ILC3 between the control and ABX groups. This implies that calculating cell numbers by immunofluorescence is not accurate.

We appreciate the reviewer's observation. The apparent increase in the numbers of ILC2s and ILC3s in scRNA-seq data following ABX treatment is not reflective of an actual rise in cell numbers but rather a result of enhanced sequencing power/slots available to detect these populations (see also point 8).

The decrease in the myeloid ratio within the CD45+ sequenced population increases the likelihood of detecting these low-frequency ILCs, leading to a higher number of detected cells. This underscores that calculating cell numbers by RNASeq for low-frequency populations when major populations shifts occur (i.e. myeloid compartment) should be considered an exploratory technique.

On the contrary, based on our data calculating absolute cell numbers by immunofluorescence for low frequency populations is a valid technique compared to scRNAseq when said population shifts happen.

We apologize for any misunderstanding arising from this aspect.

13. In Figure 6D, please provide the flow cytometry figures as well as the gating strategy.

The gating hierarchy is presented in Fig. S4C, and for a more detailed view, we have included the gating strategy here below (for Fig. 6D) and now as Fig. S4D and S4E:

Additionally, individual experiment figures captured from FlowJo screenshots are provided below, representing a total of four independent experiments comparing lamina propria leukocytes from WT and cKO mice. These screenshots illustrate the parent CD3- CD127+ and the target CD335+

population side-by-side (see below). The frequency of the target population in comparison to the Dump- CD45+ is underlined in the same color as the respective experiment. FlowJo is showing frequencies in % while in Fig.6D we depict them as fractions of 1.

Experiment 415_A

Experiment 415_B

Experiment 416_A

Experiment 421_A

14. On Line 364, please provide references, particularly for IL-15 and IL-18 promoting ILC3 expansion in the mouse intestine.

Thanks for pointing out the lack of references, we have included them now:

- “54. Victor, A. R. et al. IL-18 Drives ILC3 Proliferation and Promotes IL-22 Production via NF- κ B. *The Journal of Immunology* **199**, 2333–2342 (2017).
- 55. Robinette, M. L. et al. IL-15 sustains IL-7R-independent ILC2 and ILC3 development. *Nat Commun* **8**, 14601 (2017).”

15. On Line 380, "cxcr16" should be replaced with "cxcr6". Cxcr6 is also expressed by other T cells. Do they exhibit similar phenotypes as ILC2 or ILC3?

The typo has been rectified, and 'Cxcr16' has been corrected to 'Cxcr6'.

Regarding your inquiry about other T-cells expressing the same Cxcr6 receptor, as illustrated in Fig. 6G, we observe that Cd4+ cells also express Cxcr6 (refer to Cd4 expression distribution in Fig. S4C).

We indeed also see a modest increase in CD4+ cells in the lamina propria (see flow cytometry data below), but have not pursued that in more detail at this stage.

16. On Line 406, although there is no difference in composition of the fecal microbiota, there may be a change in its function, and also the fecal microbiota may not represent the mucosa-associated microbiota. Accordingly, it is necessary to downscale the conclusion about the microbiota-independent effect.

Indeed, we appreciate the reviewer's insightful comment. We acknowledge that, while the data from ABX-treated mice support the assertion of microbiota-independent effects due to systemic depletion, the analysis of the microbiota in icKO mice is limited to fecal samples and may not fully

represent the mucosa-associated microbiota. Consequently, we have adjusted the conclusion in that section to reflect this limitation.

17. The word "zWhile" appears to be misspelled on line 415.

Apologies, it is now fixed. Thanks for pointing that out.

18. What are the consequences of the increased ILC2 and ILC3 in LSD1 cKO mice? How does this relate to the diarrhea phenotype?

We believe, now with the new data provided, that the diarrhea is likely due to malabsorption and unrelated to ILC2/3 numbers. Indeed, we find an increase in ILC2/3s in the icKO prior to development of changes in the microbiota or development of diarrhea. Assessing the consequence of increase in ILC2/3's in this system specifically will be quite tricky, as one of their main roles include 'activating' the epithelium to induce effector responses to inflammation and infection (and it's the epithelium that is most strongly affected to begin with). As ILC2s originating from the gut can travel to the lung, we are planning follow up work to assess disease models that are not in the gut (asthma-type work). However, we feel that this is well beyond the scope of the current work.

REVIEWER COMMENTS

Reviewer #1 (Remarks to the Author):

The authors have made appropriate changes to the text and data and have highlighted some of the limitations based on my own (and other reviewer) comments. I believe it would be useful to actually write a specific "limitations of the study" subsection at the end of the discussion to provide a concise summary of the potential issues that are associated with the work to any readers. Nonetheless it is an interesting piece of work and I look forward to the authors clarifying these point of interpretation in the future.

Reviewer #2 (Remarks to the Author):

The authors have done a thorough job with the revised manuscript and answered all my questions.

Reviewer #3 (Remarks to the Author):

The authors have responded to most of my comments, and the manuscript has been improved. However, several concerns have still remained as follows.

1. In Figure 6D, ILC3 staining is unsatisfactory. And also CD3-CD127+Nkp46+ILC3 is only a subgroup of ILC3 and does not represent all ILC3s. RORgt should be used as a marker for ILC3s in flow cytometry, just as it was used for the immunofluorescence staining of intestinal sections.
2. Regarding immunofluorescence staining, it is still unclear how many sections and mice were used? How many times was the experiment repeated? Did the slices undergo a double-blind evaluation?
3. Similarly, the number of repetitions of each experiment as well as the number of mice should be clearly indicated throughout the paper. Additionally, please ensure that statistics are used appropriately throughout the paper. The authors mainly used t-test. It would be more appropriate to use a non-parametric test instead.

Nat Comms revision 2.0. Point-by-point response to the reviewers

REVIEWER COMMENTS

Reviewer #1 (Remarks to the Author):

The authors have made appropriate changes to the text and data and have highlighted some of the limitations based on my own (and other reviewer) comments. I believe it would be useful to actually write a specific "limitations of the study" subsection at the end of the discussion to provide a concise summary of the potential issues that are associated with the work to any readers. Nonetheless it is an interesting piece of work and I look forward to the authors clarifying these points of interpretation in the future.

We thank the reviewer for this suggestion, and we have now included a paragraph at the end of the discussion stating the main limitations in this study.

Reviewer #2 (Remarks to the Author):

The authors have done a thorough job with the revised manuscript and answered all my questions.

Reviewer #3 (Remarks to the Author):

The authors have responded to most of my comments, and the manuscript has been improved. However, several concerns have still remained as follows.

1. In Figure 6D, ILC3 staining is unsatisfactory. And also CD3-CD127+Nkp46+ILC3 is only a subgroup of ILC3 and does not represent all ILC3s. RORgt should be used as a marker for ILC3s in flow cytometry, just as it was used for the immunofluorescence staining of intestinal sections.

We agree with the reviewer that CD3-CD127+Nkp46+ defines a subset of ILC3s (albeit the majority). In addition, there is a possibility that ILC1s will express those markers (which intracellular Tbet and/or RORgt staining would address). That said, we feel that the proposed experiment does not alter the central messages of our study to warrant additional mouse experiments at this stage. We'd like to provide the following clarifications/arguments:

- From the scRNA-seq data, we do not observe differences in relative number of ILC1s (in either conventional or ABX treated mice, see updated Fig. S6A) – or even a slight reduction in ILC1 numbers in the conventional KO mice. Whereas ILC3s in both conventional and ABX treated had a 2-3 fold increase in KO mice from our scRNA-seq data (which is similar to the increase we observe by flow (i.e. the CD3-CD127+Nkp46+ population), and by tissue staining (i.e. the CD3-Rorgt+ population).
- Throughout the manuscript, we aim to provide an overview of interactions between epithelium and different immune cell types. For all of them, a deeper insight would be interesting (and we hope to follow up on several) – e.g. functionality of macrophage subtypes, IgA plasma cell characteristics such as longevity, 'natural' vs 'inflammatory' ILC2s, and finally the ILC3 subtypes.

Together, a deeper insight in the ILC3 population will not change the scope or main message of the study.

- Importantly; new mouse experiments and flow cytometry will actually be quite difficult and costly to perform for us at this stage. The postdoc that did the flow cytometry (Ostrop) has a new position elsewhere, and the first author also just started a new position after finalizing the last revision round (end of 2023). In addition, I have moved my lab now to Canada (all this work was done in Norway), and the Norwegian part is practically closed down. It would be quite costly to transfer/re-derive mice, my current research team would require time to gain familiarity with this study, in addition to the costs for new antibodies etc.. I hope these realities will help to shape any decisions on the absolute necessity of the suggested work.

So, what we propose is to alter the figure and the description, so that we merely focus on the markers, rather than claiming that these are all ILC3s. And include this in the new limitations of the study section at the end of the discussion.

“In addition, our approach was limited in the extent by which we could define immune cell populations. For example, there are several subsets of ILC2s and ILC3s, which we did not quantify or qualify in detail.”

Figure 6D and legend:

(D) Flow cytometry data derived from small intestinal lamina propria CD45+ cells showing (Ly6g⁻/B220⁻ CD11b⁻/CD3⁻ CD127⁺ [IL-7R α ⁺]/CD335⁺ [NKp46⁺]) frequency. Each data point represents an individual mouse, independent experiments were carried out for each WT and cKO pair, (Two-tailed unpaired t-test). Full gating hierarchy and strategy is shown in (Fig. S4C-E).

Manuscript text:

“Finally, increased numbers of Lin- CD11b- CD127+ NKp46+ (CD335) cells, a gating that largely overlaps with a predominant subset of lamina-propria ILC3s^{50,51}, was also confirmed by surface staining using flow cytometry (Fig. 6D).”

New references:

50. Satoh-Takayama N, Vosshenrich CA, Lesjean-Pottier S, Sawa S, Lochner M, Rattis F, Mention JJ, Thiam K, Cerf-Bensussan N, Mandelboim O, Eberl G, Di Santo JP. Microbial flora drives interleukin 22 production in intestinal NKp46+ cells that provide innate mucosal immune defense. Immunity. 2008 Dec 19;29(6):958-70. doi: 10.1016/j.immuni.2008.11.001. Epub 2008 Dec 11. PMID: 19084435.

51. Zeng B, Shi S, Ashworth G, Dong C, Liu J, Xing F. ILC3 function as a double-edged sword in inflammatory bowel diseases. Cell Death Dis. 2019 Apr 8;10(4):315. doi: 10.1038/s41419-019-1540-2. PMID: 30962426; PMCID: PMC6453898.

2. Regarding immunofluorescence staining, it is still unclear how many sections and mice were used? How many times was the experiment repeated? Did the slices undergo a double-blind evaluation?

We thank the reviewer for this comment, and we do apologize if our answers were not complete and clear enough in the previous round of revision.

In each figure legend, we do refer to the number of samples used per experiment. When independent rounds of experiments are carried out, this is also stated in a per figure legend basis. Each datapoint corresponds to a biological replicate (mouse), we are not plotting or comparing technical replicates. For clarity and conciseness, and as per Nature publishing group guidelines, we are additionally attaching a table where the number of biological replicates per experiment/figure are stated (see `stat_test_and_n_per_figure.xlsx`).

As for the evaluation, for each biological replicate (mouse) we extract 10cm sections of Duodenum, Jejunum, Ileum and Colon. Each of these tissue portions is processed as a swiss-roll, included in a paraffin block and assigned a number corresponding to the mouse ID.

For each block containing all 4 swiss rolls, we cut a section. So, each biological replicate is represented by a section that is stained, imaged in its entirety and quantified. Regarding quantification we do randomly select complete and intact villi across the entire swiss roll. In practice this means we count 25-30 morphologically complete villi were counted and then averaged per biological replicate. The resulting mean corresponds to 1 biological replicate data point.

Figure 1. Sample preparation for each biological replicate data point. From mouse to paraffin block.

During the entire process, from tissue collection, processing, staining, imaging and to quantification all the people involved are blind to genotype. It is not until post-quantification that the mouse ID and genotype are cross-referenced for representation and statistical testing. For treatments, either Tamoxifen, Antibiotic or aCXCL16 and their corresponding control vehicles, these are performed by qualified animal facility personnel who are also blind to mice genotype. Treatment is also cross-referenced with mouse ID post-quantification. In short, yes, slices undergo a blind evaluation.

Figure 2. During the entire process, each biological replicate is assigned a mouse ID with no mention to treatment or genotype neither in paraffin blocks nor tissue slides.

We have modified the methods section to clarify all these concerns under “Immunofluorescence staining of paraffin-embedded tissue and organoids” and “Image Acquisition, Processing Software and Quantification”:

“Image Acquisition, Processing Software and Quantification

Images were acquired using Zen 2.3 Black and grey levels/maximum intensity projections were adjusted in Zen 2.3 Blue prior to .tiff export. Cell counts were manually quantified in an unbiased and blind manner. Researchers were- blind to sample genotype and treatment during tissue collection, processing, staining, imaging and quantification- using sample numbers as identifiers. To quantify cells, we randomly selected complete and intact villi across the entire swiss roll. Based on availability, 25-30 morphologically complete villi were counted and averaged per biological replicate (one tissue section per biological replicate). After quantification each sample number was cross-referenced to genotype and treatment. Depicted scale bars apply to all images within the same panel.”

3. Similarly, the number of repetitions of each experiment as well as the number of mice should be clearly indicated throughout the paper. Additionally, please ensure that statistics are used appropriately throughout the paper. The authors mainly used t-test. It would be more appropriate to use a non-parametric test instead.

As stated in the previous answer the number of mice and repetitions are disclosed in each figure legend at the end of the manuscript or in the embedded supplementary figure legends. In addition, we are attaching a table where the number of biological replicates per experiment/figure are stated (see stat_test_and_n_per_figure.xlsx). For the sake of simplicity we are showing the requested immunofluorescences and revised statistical tests requested by Rev. 3.

Figure	Panel	NR of Biological Replicates (1 tissue section for biological replicate)						NR of independent experiments	Statistical test applied
1	A, B	7 WT (Duo)	7 cKO (Duo)	7 WT (Jej)	7 cKO (Jej)	6 WT (Ile)	7 cKO (Ile)	2	Two-tailed unpaired t-test (WT vs cKO within each tissue location)
1	A, C	7 WT (Duo)	7 cKO (Duo)	7 WT (Jej)	7 cKO (Jej)	6 WT (Ile)	7 cKO (Ile)	2	Two-tailed unpaired t-test (WT vs cKO within each tissue location)
1	G	5 WT (Duo)	5 cKO (Duo)	5 WT (Jej)	5 cKO (Jej)	5 WT (Ile)	5 cKO (Ile)	2	Representative image, no quantification.
1	L	15 WT	15 cKO					4	Contingency analysis by Fisher's exact test
1	K	12 WT	12 cKO					4	Two-tailed unpaired t-test (WT vs cKO within each timepoint)
51	B	3 WT	3 cKO					1	Two-tailed unpaired t-test (WT vs cKO within each tissue location)
51	F	5 WT (Duo)	5 cKO (Duo)	4 WT (Jej)	5 cKO (Jej)	4 WT (Ile)	5 cKO (Ile)	1	Two-tailed Mann-Whitney non-parametric test for Duodenum and Two-tailed t test for Jejunum and Ileum (WT vs cKO)
51	G	3 WT	3 cKO					1	Two-tailed unpaired t-test (WT vs cKO within each tissue location)
2	D	5 WT (Duo, Jej, Ile)	5 cKO (Duo, Jej, Ile)	3 WT (Colon)	3 cKO (Colon)			2	Representative image, no quantification.
3	B	10 WT	10 cKO					3	Two-tailed unpaired t-test (WT vs cKO)
53	A	6 WT (Ctrl)	3 cKO (Ctrl)	9 WT (ABX)	6 cKO (ABX)			3	One-tailed unpaired t-test (Control vs ABX within genotype)
53	G	3 WT (Ctrl)	3 cKO (Ctrl)	3 WT (ABX)	3 cKO (ABX)			2	Two-tailed unpaired t-test (Control vs TAM within genotype)
53	H	3 WT (Ctrl)	3 cKO (Ctrl)	3 WT (ABX)	3 cKO (ABX)			2	One-tailed Mann-Whitney non-parametric test
4	B	5 WT	5 cKO					4	Two-tailed Mann-Whitney non-parametric test
4	C	4 WT	4 cKO					4	Two-tailed unpaired t-test (WT vs cKO)
5	C	4 WT (Ctrl)	4 cKO (Ctrl)	10 WT (ABX)	10 cKO (ABX)			3	One-tailed Mann-Whitney non-parametric test (WT vs cKO within treatments)
5	E, F	5 WT	5 cKO					2	Two-tailed unpaired t-test (WT vs cKO)
6	A, B	6 WT (Ctrl, Duo)	6 cKO (Ctrl, Duo)	5 WT (ABX, Duo)	5 cKO (ABX, Duo)			2	Two-tailed unpaired t-test (WT vs cKO within treatments)
6	A, C	7 WT (Ctrl, Duo)	7 cKO (Ctrl, Duo)	4 WT (ABX, Duo)	5 cKO (ABX, Duo)			2	Two-tailed Mann-Whitney non-parametric test (WT vs cKO within treatments)
6	D	4 WT	4 cKO					4	Two-tailed unpaired t-test (WT vs cKO)
56	B	7 WT (Duo)	7 cKO (Duo)	6 WT (Jej)	5 cKO (Jej)	7 WT (Ile)	5 cKO (Ile)	2	Two-tailed Mann-Whitney non-parametric test for Duodenum and Two-tailed t test for Jejunum and Ileum (WT vs cKO)
56	B	4 WT (Duo)	5 cKO (Duo)	5 WT (Jej)	5 cKO (Jej)	5 WT (Ile)	4 cKO (Ile)	2	Two-tailed unpaired t-test (WT vs cKO within each tissue location)
7	A, B	4 mice per genotype	WT and cKO) per timepoint (0, 7, 14, 28, 42 days)					1	Two-tailed unpaired t-test (WT vs cKO within each timepoint)
7	C, D	4 WT (isotype)	3 cKO (isotype)	4 WT (aCKCL16)	3 cKO (aCKCL16)			1	Two-tailed unpaired t-test (Control vs aCKCL16 within genotype)
7	C, E	4 WT (isotype)	3 cKO (isotype)	4 WT (aCKCL16)	3 cKO (aCKCL16)			1	Two-tailed unpaired t-test (Control vs aCKCL16 within genotype)

Figure 3. See *stat_test_and_n_per_figure.xlsx*. Sample replicates and associated statistical analysis applied in a per figure basis.

Furthermore, we double-checked our data for Gaussian/normal distribution along the paper applying the Kolmogorov-Smirnov and Shapiro-Wilk tests for normality. We thank the reviewer for this comment, and we have now applied Mann-Whitney, one- or two-tailed, nonparametric test for non-Gaussian distributions when needed (Fig. S1F, Fig. S3H, Fig. 4B, Fig.5C, Fig. 6C and Fig. S6B).

These new analyses did not change the statistical significance of the data. The degree of significance however has changed for Fig. S3H and Fig. 6C. We have corrected these figures and indicated the used test for each analysis in the corresponding figure legends. We have also updated the corresponding Methods section:

“Statistical Analysis

*Unless stated otherwise in the methods section, statistical analysis and representation was carried out using Graphpad Prism V8.0. Prior to determining statistical significance, we assessed data distribution using the Kolmogorov-Smirnov (KS) and Shapiro-Wilk (SW) tests. In instances where the sample size was too small to confidently apply the KS test, the SW test was exclusively used. Normality was affirmed only when data passed both tests (where applicable), allowing for the subsequent application of parametric tests to analyze statistical significance. Conversely, for data that did not meet these criteria, non-parametric tests were utilized (One- or Two-tailed Mann-Whitney). We excluded RNASeq data expression and microbial population analyses, as these datasets incorporate inherent checks for data distribution within their analysis pipelines. Statistical significance was evaluated by the appropriate methods stated in the figure legends or in methodology section. Means and individual data points (biological replicates) were represented as \pm SEM unless stated otherwise. Differences were considered as statistically significant at $*P<0.05$, $**P<0.01$, $***P<0.001$, and $****P<0.0001$. We have removed statistical significance from comparisons between the villus compartments for crypt-restricted markers like Mki67, Lgr5, Olfm4, Lyz, and Defa22, and vice versa for Slc15a1, Cd36, Apob and Npc1l1, marking them as "n/a" (does not apply) to avoid potential data misinterpretation.”*

For the sake of thoroughness, we have also applied non-parametric testing even for small sample sizes that passed the normality test. In none of the occasions the statistical significance was lost.

REVIEWERS' COMMENTS

Reviewer #3 (Remarks to the Author):

The authors have addressed all my comments, and the manuscript has been significantly improved.